# WBMM: Windowed Batch Matrix Multiplication for Efficient Large Receptive Field Convolution

Wan Song[1]  Wei Zhou[2]  Rui Wang[2]  Jun Yu[3]  Toru Kurihara[3]  Jiajia Xu[2]  Shu Zhan[1]

## Abstract

Large kernel depthwise convolutions achieve strong performance but suffer from significant degradation as kernel size grows due to irregular memory access from gather-based computation; while Large Kernel Acceleration (LKA) helps on small feature maps, it becomes counterproductive on large feature maps, even slower than non-accelerated implementations. We propose Windowed Batch Matrix Multiplication (WBMM), which *partitions* input into contiguous windows and *indexes* a compact relative position bias table to construct weight matrices, enabling regular memory access via batched matrix multiplication. This yields a unique property: WBMM's throughput improves with larger windows, opposite to depthwise convolutions that degrade with larger kernels. Operator-level benchmarks show WBMM with $14 \times 14$ windows outperforms $5 \times 5$ depthwise convolution baselines in speed while providing a $7.8\times$ larger per-layer receptive field. Combined with inter-block cross-window communication and hierarchical window reparameterization, WBMM achieves comparable or higher accuracy on ImageNet-1K, COCO, and ADE20K with $1.31$–$1.88\times$ training speedup, and demonstrates consistent advantages across GPU, CPU, and edge devices without requiring specialized acceleration kernels. Our code is available at https://github.com/wansong-s/WBMM.

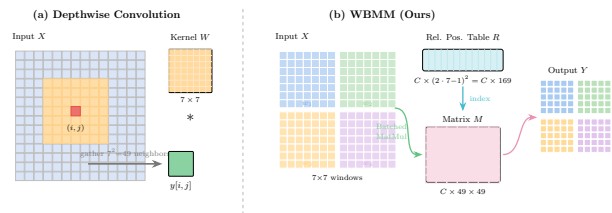

*Figure 1.* **Depthwise convolution vs. WBMM.** (a) Depthwise convolution gathers $k^2$ scattered neighbors per output, causing irregular memory access that worsens with kernel size. (b) WBMM partitions input into contiguous windows and constructs weights via table indexing, enabling regular memory access through batched matrix multiplication.

## 1. Introduction

Convolutional Neural Networks (CNNs) have undergone significant architectural evolution. While early designs relied on stacking small kernels (typically $3 \times 3$), recent studies have demonstrated substantial gains from large convolution kernels. This shift was driven by ConvNeXt (Liu et al., 2022) with $7 \times 7$ kernels and further pushed by RepLKNet (Ding et al., 2022b) ($31 \times 31$ kernels), SLaK (Liu et al., 2023) ($51 \times 51$ kernels), and UniRepLKNet (Ding et al., 2024) ($13 \times 13$ kernels).

However, large kernel convolutions face fundamental computational challenges that severely limit their practical deployment. Standard convolution implementations must gather scattered input neighborhoods for each output position, causing increasingly irregular memory access patterns as kernel size grows.

**The core problem.** Our comprehensive operator-level benchmarks (Section 4.1) reveal that standard depthwise convolutions slow down by **71–78%** when kernel size increases from $5 \times 5$ to $13 \times 13$, and by **92–94%** for $27 \times 27$ kernels. This degradation is consistent across all batch sizes (4, 16, 64, 128, 256) and feature map resolutions ($\geq 28 \times 28$), demonstrating that the performance penalty is *inherent* to the memory access pattern rather than implementation-specific.

Figure 1 contrasts the two computation paradigms at a high level. As Figure 2 illustrates in more detail, the root cause is

[1]Hefei University of Technology, Hefei, China [2]Lingyang Industrial Internet Co., Ltd., Hefei, China [3]Kochi University of Technology, Kochi, Japan. Correspondence to: Shu Zhan <shu_zhan@hfut.edu.cn>, Jiajia Xu <xujiajia@mail.ustc.edu.cn>.

*Proceedings of the 43rd International Conference on Machine Learning*, Seoul, South Korea. PMLR 306, 2026. Copyright 2026 by the author(s).

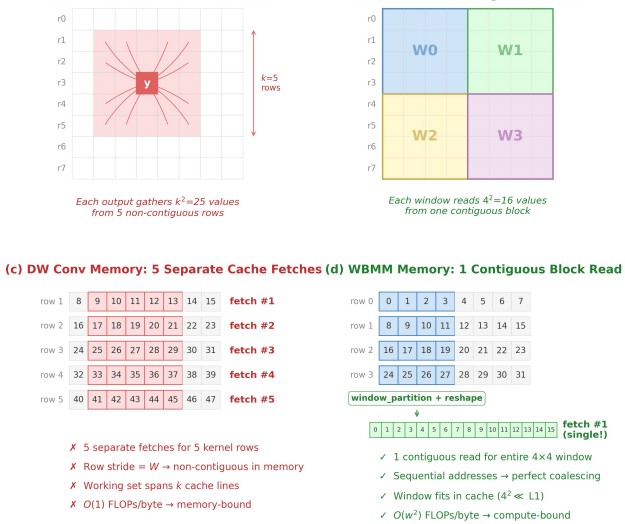

*Figure 2.* **GPU memory access pattern: depthwise convolution vs. WBMM.** (a,c) A $5 \times 5$ depthwise convolution gathers 25 values from 5 non-contiguous rows, requiring 5 separate cache fetches with stride $W$. (b,d) WBMM reads each window as a single contiguous block, fitting entirely in L1 cache and enabling coalesced access. A $4 \times 4$ window is shown here for illustration only; actual WBMM configurations use $7 \times 7$ or $14 \times 14$ windows.

that gather-based depthwise convolution is memory-bound: each output collects $k^2$ values from non-contiguous rows, exceeding L1/L2 cache and yielding only $O(1)$ FLOPs per loaded element. WBMM eliminates this irregularity by reading each window as one contiguous block, shifting the operator from memory-bound to compute-bound (full arithmetic-intensity analysis in Section 3.4).

**The limitation of existing acceleration.** While the Large Kernel Acceleration (LKA) CUDA kernels from RepLKNet/UniRepLKNet mitigate this degradation on small feature maps during training, their fixed tiling no longer matches large feature maps, where they become **even slower than the non-accelerated baseline**. At $224 \times 224$ resolution and batch=128, LKA with $7 \times 7$ kernels is **80% slower** than baseline, and $13 \times 13$ kernels are **89% slower**. This severely limits deployment of large kernels in tasks that process high-resolution feature maps, such as semantic segmentation, object detection, and super-resolution.

### 1.1. Our Approach: Traversing Parameters Instead of Data

We propose Windowed Batch Matrix Multiplication (WBMM), which changes the computation paradigm by *traversing the parameter table* instead of input data. As illustrated in Figure 1:

- **Depthwise convolution**: For each output position,

*gather* $k^2$ scattered input values → irregular memory access.

- **WBMM**: *Partition* input into contiguous windows, *index* parameter table to construct weight matrix → regular memory access.

Concretely, WBMM partitions input into contiguous non-overlapping windows (e.g., $7 \times 7$) and constructs a weight matrix $M \in \mathbb{R}^{C \times 49 \times 49}$ by indexing into a compact relative position bias table $R \in \mathbb{R}^{C \times 169}$; the output is computed via batched matrix multiplication on contiguous memory blocks. This "partition-then-multiply" design is why WBMM's efficiency **does not degrade** as feature map resolution or window size grows—the batched matrix multiplication fully leverages GPU parallelism and actually improves with more windows.

### 1.2. Contributions

Our main contributions are:

- **Novel computation paradigm**: We replace the gather-based memory access of large-kernel depthwise convolution with a "partition-then-index" scheme: WBMM indexes a compact relative-position bias table to form a per-channel weight matrix, then applies it via batched matrix multiplication on contiguous windows. Throughput *improves* with larger windows and stays consistent across resolutions—versus the 80–89% LKA slowdown at $224 \times 224$ (batch=128)—making WBMM suitable for classification, detection, segmentation, and super-resolution.

- **Batch-independent weight construction**: Unlike window attention, which builds an input-dependent matrix per sample and window ($O(B \cdot N_h N_w \cdot d^2)$), WBMM constructs a *single* channel-dependent matrix ($O(C \cdot d^2)$), scaling better with batch size and resolution.

- **Three lightweight components**: (i) inter-block $3 \times 3$ depthwise convolutions for cross-window communication, reaching comparable or higher accuracy at higher speed than intra-block alternatives; (ii) inference-time weight-matrix caching that removes redundant indexing on small feature maps; and (iii) hierarchical window reparameterization, a dual-scale fusion improving dense-prediction accuracy without slowing inference.

- **Hardware-agnostic efficiency**: Without specialized kernels, WBMM is consistently faster on GPU, CPU, and edge devices (Section K); across ImageNet-1K, COCO, and ADE20K it delivers a 1.31–1.88× training speedup and, with hierarchical reparameterization, surpasses UniRepLKNet in both mIoU and AP.

## 2. Related Work

### 2.1. Large Convolution Kernels in CNNs

The resurgence of large convolution kernels began with ConvNeXt (Liu et al., 2022), which showed that $7 \times 7$ depthwise convolutions could match Transformer performance. This trend continued with RepLKNet (Ding et al., 2022b), which extended kernels to $31 \times 31$. SLaK (Liu et al., 2023) reduced parameters through sparse rectangular decomposition, reaching $51 \times 51$ kernels, though with suboptimal computational efficiency. PeLK (Chen et al., 2024) achieved linear parameter scaling through parameter sharing, enabling $101 \times 101$ kernels but still faced quadratic computational complexity in $k$.

UniRepLKNet (Ding et al., 2024) reported that $13 \times 13$ is a strong kernel size across various modalities, based on extensive ablations that balance performance and computational cost. However, its structural reparameterization adds training overhead and relies on specialized GPU kernels that can become counterproductive during inference on large feature maps.

### 2.2. Convolution Implementations and Their Limitations

**Im2col+GEMM.** The classic im2col+GEMM approach (Chellapilla et al., 2006; Jia et al., 2014) converts convolution to matrix multiplication by extracting each $k \times k$ receptive field as columns. For input $X \in \mathbb{R}^{B \times C \times H \times W}$, this creates an expanded matrix of size $\mathbb{R}^{BC \times HW \times k^2}$. The $k^2$ memory expansion becomes prohibitive for large kernels, and gathering scattered neighbors creates irregular memory access that worsens with kernel size.

**Standard Depthwise Convolution (DW-Std).** Modern deep learning frameworks like PyTorch (Paszke et al., 2019) use implicit GEMM via cuDNN (Chetlur et al., 2014) for depthwise convolutions. While implicit GEMM avoids explicit im2col materialization by computing indices on-the-fly, reducing memory overhead from $O(k^2)$ to $O(1)$, the fundamental limitation remains the *gather-based* computation pattern: for each output position, the implementation must compute indices to gather $k^2$ scattered input values. As kernel size increases, these memory accesses become increasingly non-coalesced, causing the 71–78% degradation we observe from $5 \times 5$ to $13 \times 13$ kernels.

**Large Kernel Acceleration (LKA).** RepLKNet (Ding et al., 2022b) and UniRepLKNet (Ding et al., 2024) provide specialized CUDA kernels (we call these LKA throughout, distinct from VAN's Large Kernel Attention (Guo et al., 2023)) that tile the depthwise convolution so a working block stays in on-chip shared memory, improving data reuse. The gather-based access pattern itself is unchanged—each

output still collects $(2k_h + 1)(2k_w + 1)$ scattered inputs— so this reuse holds only while the block stays on-chip, i.e. on small feature maps ($\leq 14 \times 14$). On larger maps its fixed compile-time tiling no longer matches the workload and, lacking cuDNN's adaptive algorithm selection, the kernel runs *slower* than the standard baseline—likely a fixed-configuration effect: at $224 \times 224$ and batch=128, LKA with $13 \times 13$ kernels is **89% slower** than DW-Std $5 \times 5$ (Table 13).

**WBMM (Ours).** Unlike all above methods that traverse input data to gather scattered neighborhoods, WBMM traverses only the parameter table; input data remains in contiguous window blocks, providing regular memory access regardless of effective kernel size or feature map resolution. The arithmetic-intensity analysis explaining *why* this enables acceleration with larger windows is deferred to Section 3.4.

### 2.3. Window-Based Processing in Vision Models

Window-based processing has become a key efficiency technique in vision models. Vision Transformer (ViT) (Dosovitskiy et al., 2021) successfully adapts transformers to images, while Swin Transformer (Liu et al., 2021) introduces efficient window-based self-attention with non-overlapping windows. For cross-window information exchange, Swin employs shifted window partitioning in alternate layers, and has been applied to various tasks (Liang et al., 2021; Hatamizadeh et al., 2021; Cao et al., 2022).

**Critical distinction from window attention.** Unlike Swin-style window attention that computes *input-dependent* matrices and applies a softmax per sample/window, WBMM constructs a single *input-independent* weight matrix $M$ shared across the whole batch; $M$ can be cached at inference (Section 3.4.5) and tiled with a simple purely-linear kernel. We empirically compare against a parameter-matched non-overlapping window attention in Section D.

### 2.4. Structural Reparameterization

Structural reparameterization enhances neural networks by adding branches during training that are merged at inference. RepVGG (Ding et al., 2021) pioneered this approach by adding $1 \times 1$ convolutions and identity shortcuts during training and fusing them at inference. ACNet (Ding et al., 2019) incorporated asymmetric convolutions, while RepLKNet (Ding et al., 2022b) transformed large $31 \times 31$ kernels into functionally equivalent smaller $5 \times 5$ kernels. RepMLPNet (Ding et al., 2022a) introduced locality injection for fully-connected layers.

Our WBMM method introduces a structural reparameterization tailored to window-based batch matrix multiplication with embedded position bias. We apply this minimal repa-

rameterization only to the smallest model variants (WBMM-P and WBMM-N) at the final stage (S4) for image classification tasks, avoiding the heavy multi-branch training overhead of UniRepLKNet.

## 2.5. Position Encoding in Vision Networks

Position encoding is crucial for modeling spatial relationships. Existing approaches include absolute position encoding (Dosovitskiy et al., 2021), relative position bias (Liu et al., 2021; Shaw et al., 2018), and conditional position encoding (Chu et al., 2023). Unlike Swin (Liu et al., 2021), which shares a single relative position bias across all samples, heads, and windows, our approach adopts channel-specific position-aware weights, enhancing representational capacity while maintaining computational efficiency.

## 3. Method

### 3.1. Notation and Definitions

We establish notation used throughout this paper: $X \in \mathbb{R}^{B \times C \times H \times W}$ denotes the input feature map (batch $B$, channels $C$, spatial $H \times W$); $(h, w)$ are spatial position indices, not to be confused with the window size $w_h \times w_w$ or the feature-map width $W$; $N = B \cdot N_h \cdot N_w$ is the total number of windows, where $N_h = H/w_h$, $N_w = W/w_w$; $d = w_h \cdot w_w$ is positions per window; $R \in \mathbb{R}^{C \times (2w_h-1)(2w_w-1)}$ is the relative position bias table; $I \in \mathbb{Z}^{d \times d}$ is the relative position index matrix; and $M \in \mathbb{R}^{C \times d \times d}$ is the weight matrix constructed from $R$. To avoid clash between $W$ (feature-map width) and depthwise kernel weights, we denote the kernel by $K_{c,p,q}$ in Equation (1) below; likewise, to avoid clash with the batch index $b$, we write the per-channel bias as $\beta_c$.

### 3.2. Problem Formulation: Why Large Kernels Are Slow

Standard depthwise convolution with kernel size $(2k_h + 1) \times (2k_w + 1)$ computes:

$$Y_{b,c,h,w} = \sum_{i=-k_h}^{k_h} \sum_{j=-k_w}^{k_w} K_{c,\,i+k_h,\,j+k_w} \cdot X_{b,c,h+i,w+j} + \beta_c \tag{1}$$

where $K_c$ is the depthwise kernel for channel $c$ and $\beta_c$ is a per-channel bias.

The key computational challenge is that for each output position $(h, w)$, the implementation must *gather* $(2k_h + 1)(2k_w + 1)$ input values $X_{b,c,h+i,w+j}$ from scattered memory locations. As kernel size $k$ increases, these memory accesses become increasingly non-coalesced, causing severe performance degradation regardless of the underlying implementation strategy.

### 3.3. Theoretical Foundation: Convolution–Matrix Equivalence

We establish that depthwise convolutions on bounded feature maps can be exactly represented as matrix multiplication with position-dependent weights.

**Theorem 3.1** (Maximum Effective Kernel Size). *For a feature map with spatial dimensions $H \times W$, the maximum effective kernel size that covers all valid relative offsets is $(2H - 1) \times (2W - 1)$.*

*Proof.* For any two positions $(h_1, w_1)$ and $(h_2, w_2)$ in an $H \times W$ feature map, the relative offset $(\delta_h, \delta_w) = (h_1 - h_2, w_1 - w_2)$ satisfies $\delta_h \in [-(H-1), H-1]$ and $\delta_w \in [-(W-1), W-1]$, spanning exactly $(2H-1) \times (2W-1)$ distinct relative positions. $\square$

**Theorem 3.2** (Convolution–Matrix Equivalence). *For a feature map with spatial dimensions $H \times W$ and any depthwise convolution with kernel size $(2k_h + 1) \times (2k_w + 1)$ (with the standard zero-padding convention on the input), the convolution can be exactly represented as*

$$y_c = x_c M_c + \beta_c \mathbf{1}^\top, \tag{2}$$

*where $x_c \in \mathbb{R}^{B \times HW}$ is the flattened input for channel $c$, $M_c \in \mathbb{R}^{HW \times HW}$ is a position-dependent weight matrix whose entries depend only on the relative offset between input and output positions, and $\mathbf{1} \in \mathbb{R}^{HW}$ broadcasts the scalar bias $\beta_c$ to every output position.*

*Proof.* We adopt the standard zero-padding convention $X_{b,c,m,n} = 0$ for $(m,n) \notin [0, H-1] \times [0, W-1]$, and extend the kernel by $K_{c,p,q} = 0$ for $(p,q) \notin [0, 2k_h] \times [0, 2k_w]$. We then construct $M_c$ in three steps.

*Step 1: Variable substitution with safe range extension.* In Equation (1), substitute $m = h + i$, $n = w + j$. The original sum over $(i,j) \in [-k_h, k_h] \times [-k_w, k_w]$ becomes a sum over $(m,n) \in [h-k_h, h+k_h] \times [w-k_w, w+k_w]$. We extend the index range to $[0, H-1] \times [0, W-1]$. By the two extensions above, every added term satisfies either $K_{c,\,m-h+k_h,\,n-w+k_w} = 0$ (when $|m-h| > k_h$ or $|n-w| > k_w$) or $X_{b,c,m,n} = 0$ (when $(m,n)$ is outside the feature map), so the added contributions vanish identically and the sum value is preserved:

$$Y_{b,c,h,w} = \sum_{m=0}^{H-1} \sum_{n=0}^{W-1} K_{c,\,m-h+k_h,\,n-w+k_w} \cdot X_{b,c,m,n} + \beta_c. \tag{3}$$

*Step 2: Index linearization.* Linearize the 2D coordinates by the bijection $t_1 = mW + n$, $t_2 = hW + w$ between $[0, H-1] \times [0, W-1]$ and $[0, HW-1]$. This unique encoding yields a matrix $M_c \in \mathbb{R}^{HW \times HW}$. The linearization handles any kernel size: for kernels larger than

$(2H-1) \times (2W-1)$, excess parameters multiply only zero inputs (by Theorem 3.1) and receive zero gradient, so they do not contribute to the output; for smaller kernels, excess matrix entries are zero.

*Step 3: Offset-dependent entry definition.* For each pair $(t_1, t_2)$, recover $(m, n)$ and $(h, w)$ via the inverse of the linearization and set $\delta_h = m - h$, $\delta_w = n - w$. Define

$$M_c[t_1, t_2] = \begin{cases} K_{c, \delta_h + k_h, \delta_w + k_w} & \text{if } |\delta_h| \leq k_h, |\delta_w| \leq k_w, \\ 0 & \text{otherwise.} \end{cases}$$

Substituting this into the matrix product $(x_c M_c)_{b, t_2} = \sum_{t_1} X_{b, c, m(t_1), n(t_1)} M_c[t_1, t_2]$ reproduces Equation (3), and hence Equation (1), for every $(b, c, h, w)$.

Crucially, $M_c[t_1, t_2]$ is determined entirely by the relative offset $(\delta_h, \delta_w)$: any two index pairs with the same $(\delta_h, \delta_w)$ receive identical values. The matrix $M_c$ thus possesses a block-Toeplitz structure—a property WBMM exploits via a compact bias table $R$ of size $O((2w_h-1)(2w_w-1))$ per channel rather than a dense $HW \times HW$ matrix. $\square$

**Design choice: from exact equivalence to a windowed approximation.** Theorem 3.2 is exact but impractical to instantiate globally: at $56 \times 56$, $M_c$ would carry $HW = 3{,}136$ rows and $\sim 9.8$M entries per channel. Following the locality strategy of Swin Transformer (Liu et al., 2021), we apply Equation (2) *inside* non-overlapping $w_h \times w_w$ windows, which shrinks each matrix to $\mathbb{R}^{d \times d}$ with $d = w_h w_w$ (e.g., $49 \times 49$ for $7 \times 7$ windows). WBMM is therefore not an exact realization of global convolution, but a windowed design *inspired* by the equivalence. The three subsequent components each address a trade-off introduced by this restriction: inter-block $3 \times 3$ depthwise mixing restores cross-window connectivity (Section 3.5); hierarchical reparameterization recovers multi-scale context (Section 3.6); and the input-independence of $M$ enables zero-overhead inference caching (Section 3.4.5).

### 3.4. Windowed Batch Matrix Multiplication

Building on the equivalence theorem and the above design choice, we design WBMM to efficiently process arbitrary-sized feature maps through window partitioning and relative position indexing. Algorithm 1 presents the complete algorithm.

#### 3.4.1. HANDLING ARBITRARY INPUT SIZES

A key advantage of WBMM is its ability to handle **arbitrary input resolutions** through zero-padding. For any input with spatial dimensions $H \times W$, we compute the required

---

**Algorithm 1** WBMM Forward Pass with Optional Caching

---

**Require:** Input $X \in \mathbb{R}^{B \times C \times H \times W}$, window size $(w_h, w_w)$
**Require:** Learnable parameters: $R \in \mathbb{R}^{C \times (2w_h-1)(2w_w-1)}$
**Require:** Precomputed buffer: $I \in \mathbb{Z}^{d \times d}$
**Require:** Flag: `use_cache`
 1: **// Step 1: Window Partitioning with Zero-Padding**
 2: Pad $X$ to dimensions divisible by $(w_h, w_w)$ if needed
 3: Partition into $N_h \times N_w$ non-overlapping windows
 4: $X_{\text{batch}} \leftarrow$ permute_and_reshape$(X) \in \mathbb{R}^{C \times N \times d}$, where $N = B \cdot N_h \cdot N_w$   *// $C$ is placed first so that Py-Torch* `bmm` *treats channels as the batch axis and shares $M_c$ across all $N$ windows of channel $c$*
 5: **// Step 2: Construct Weight Matrix**
 6: **if** `use_cache` **and** $M_{\text{cached}}$ exists **then**
 7:    $M \leftarrow M_{\text{cached}}$
 8: **else**
 9:    $M \leftarrow R[:, I.\text{flatten}()].\text{view}(C, d, d)$
10:    **if** `use_cache` **then**
11:       $M_{\text{cached}} \leftarrow M$
12:    **end if**
13: **end if**
14: **// Step 3: Batch Matrix Multiplication**
15: $Y_{\text{batch}} \leftarrow X_{\text{batch}} \cdot M$
16: **// Step 4: Inverse Reshape**
17: $Y \leftarrow$ inverse_reshape$(Y_{\text{batch}}) \in \mathbb{R}^{B \times C \times H \times W}$
18: **return** $Y$

---

padding as:

$$\text{pad}_h = (w_h - H \bmod w_h) \bmod w_h, \tag{4}$$
$$\text{pad}_w = (w_w - W \bmod w_w) \bmod w_w. \tag{5}$$

This ensures the padded dimensions $H' = H + \text{pad}_h$ and $W' = W + \text{pad}_w$ are divisible by the window size. After processing, the padding is removed to restore the original output dimensions. Section G reports an empirical comparison against reflection and replication padding, showing that zero-padding is the most accurate among the three.

#### 3.4.2. RELATIVE POSITION BIAS TABLE AND INDEXING

The core of WBMM is the *relative position bias table* $R \in \mathbb{R}^{C \times (2w_h-1)(2w_w-1)}$, storing learnable weights for all unique relative offsets within a window.

We precompute the *relative position index matrix* $I \in \mathbb{Z}^{d \times d}$:

$$\delta_h = h_i - h_j, \quad \delta_w = w_i - w_j, \tag{6}$$
$$I[i, j] = (\delta_h + w_h - 1) \cdot (2w_w - 1) + (\delta_w + w_w - 1). \tag{7}$$

The weight matrix is constructed via index selection:

$$M = R[:, I.\text{flatten}()].\text{view}(C, d, d). \tag{8}$$

### 3.4.3. WHY WBMM ACCELERATES WITH LARGER WINDOWS

A critical finding is that WBMM *speeds up* when window size increases from $7 \times 7$ to $14 \times 14$—the *opposite* behavior to depthwise convolutions. The reason lies in arithmetic intensity. For a window of size $w \times w$: data reads are $O(w^2)$ elements per window, computation is $O(w^4)$ multiply-add operations, yielding arithmetic intensity of $O(w^2)$ FLOPs per loaded element (the single weight matrix $M$ is shared across all $N$ windows in the batch, so its $O(w^4)$ load cost is amortized over $N$ windows and becomes negligible relative to the per-window input read of $O(w^2)$; hence the per-window arithmetic intensity is dominated by input reads). Larger windows provide higher arithmetic intensity, allowing GPU compute units to be more fully utilized.

### 3.4.4. WHY WBMM SCALES WITH FEATURE MAP SIZE

Unlike LKA-accelerated convolutions whose fixed tiling no longer matches large feature maps, WBMM's efficiency *improves* with larger feature maps because: (1) more windows provide better GPU occupancy and parallelism; (2) the batched matrix multiplication amortizes fixed overhead across more windows; (3) memory access remains regular regardless of total feature map size.

### 3.4.5. INFERENCE CACHING MECHANISM

During training, $M$ must be reconstructed at each forward pass because gradients flow through $R$. During inference, $M$ is constant and can be computed once and cached, providing additional speedup on small feature maps where index operation overhead is relatively higher. We refer to the two modes as **WBMM-NC** (no-cache; $M$ reconstructed each forward pass, used in training) and **WBMM-C** (cached; $M$ computed once at inference); they produce identical outputs and differ only in whether $M$ is recomputed.

### 3.5. Inter-Block Cross-Window Communication

Non-overlapping windows cannot directly exchange information. We address this by placing lightweight $3 \times 3$ depthwise convolutions as *separate blocks* between WBMM blocks in an **inter-block** design (schematically, Block$_n$ = WBMM, Block$_{n+1}$ = DWConv$_{3\times3}$); the actual stage-wise patterns used in our models (e.g., [W,D,W], [W↔D]×9, [W,W,W]) are task-specific and listed in Table 20. How densely these $3 \times 3$ blocks are interleaved with WBMM, and at which stages, is task-specific; we determine the patterns empirically in Section 4.2.2. Keeping WBMM and $3 \times 3$ convolutions in *separate* blocks—rather than pairing a $3 \times 3$ with WBMM inside every block—lets each operator run as a clean, highly optimized kernel, improving training and inference speed through more regular memory access and better operator fusion while retaining strong accuracy.

### 3.6. Hierarchical Window Reparameterization

For dense prediction tasks requiring multi-scale context, we introduce hierarchical window reparameterization combining global and local window processing (abbreviated "Hier" in tables; equivalently "G+L" in Table 4). The detailed algorithm is in Section N.

**Dual-Scale Bias Tables.** We maintain two learnable tables: $R_g \in \mathbb{R}^{C \times (2w_g - 1)^2}$ for large windows and $R_l \in \mathbb{R}^{C \times (2w_l - 1)^2}$ for smaller sub-windows, where $w_g = 2w_l$.

**Weight Fusion.** Each global window decomposes into a $2 \times 2$ grid of local sub-windows, indexed by $s \in \{0, 1, 2, 3\}$. During training the two scales are processed independently, each with an identity shortcut; at inference they collapse into a single matrix that adds the local pattern $M_l$ to the four diagonal blocks of the global matrix $M_g$:

$$M_{\text{fused}}[c, B_{s,s}] = M_g[c, B_{s,s}] + M_l[c], \quad s \in \{0, 1, 2, 3\}, \tag{9}$$

while the off-diagonal cross-sub-window blocks are left to $M_g$. The residual shortcuts are absorbed as identity terms in the fused matrix; the exact form and indexing are given in Section N. The fusion is a one-time tensor addition performed when the model is loaded, after which $M_{\text{fused}}$ is cached; hierarchical reparameterization therefore adds no runtime cost over single-scale WBMM *at the same window size* ($w = 14$).

### 3.7. Multi-Kernel Fusion for Minimal Feature Maps

For the S4 stage of Pico and Nano variants processing $7 \times 7$ feature maps, we introduce parallel depthwise convolution paths fused with the WBMM main branch: $Y = \text{WBMM}(X) + \text{BN}_1(\text{DW}_5(X)) + \text{BN}_2(\text{DW}_3(X))$, where $\text{DW}_k$ denotes kernel size $k \times k$. At inference, the parallel DW paths are fused into the WBMM weight matrix $M$, achieving a 0.2 percentage point improvement in Top-1 over single-path WBMM at zero additional inference cost.

## 4. Experiments

We conduct experiments on ImageNet-1K (Deng et al., 2009) classification, ADE20K (Zhou et al., 2019) semantic segmentation with UPerNet (Xiao et al., 2018), and COCO (Lin et al., 2014) object detection with Cascade Mask R-CNN (Cai & Vasconcelos, 2019).

**Implementation details.** All experiments use **FP32 precision**; training runs on **8× NVIDIA A800 GPUs** (80GB) and inference benchmarks on a **single A800** unless otherwise specified. The implementation follows the UniRepLKNet codebase with identical training protocols, hyperparameters, and data preprocessing; our only changes are replacing large kernels with WBMM and adding inter-block mixing. We use 300-epoch ImageNet training with AdamW (Loshchilov

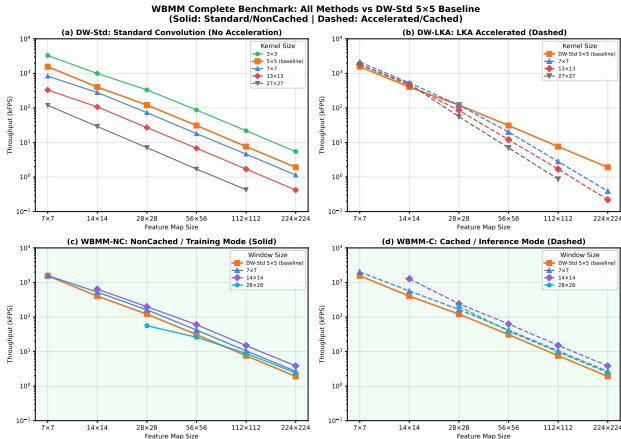

*Figure 3.* **Operator-level benchmark** (batch=128, 256 channels, FP32, single A800 GPU). DW-Std $5 \times 5$ serves as baseline. See text for detailed analysis.

& Hutter, 2019), 160k iterations for ADE20K, and a $3\times$ schedule for COCO. Ablation metrics are mean±std over three runs; full configurations are in Section L.

**Why compare with UniRepLKNet.** UniRepLKNet (Ding et al., 2024) identified $13 \times 13$ as the *optimal* kernel size through extensive cross-modal ablations, making it a rigorous state-of-the-art baseline for validating WBMM. Comparisons against further baselines—SLaK-style decomposition, non-overlapping window attention, and a ConvNeXt-T drop-in replacement—are in Section C.

### 4.1. Comprehensive Operator-Level Benchmark

To validate WBMM's efficiency, we conduct operator-level benchmarks on **single-layer feature maps**. This isolated setup makes the conclusions *architecture-agnostic*: the speedup characteristics hold regardless of whether WBMM is deployed in UniRepLKNet, ConvNeXt, or other CNN architectures.

All measurements use 256 channels, FP32 precision on a single NVIDIA A800 GPU (80GB), with 5 independent runs and IQR-based outlier removal. We present batch=128 results here; complete results across all batch sizes (4, 16, 64, 128, 256) are provided in Section I.

#### 4.1.1. KEY FINDINGS FROM BENCHMARK ANALYSIS

Figure 3 presents the comprehensive benchmark results across four configurations (batch=128), revealing several important findings.

**Finding 1: Standard Convolution Degrades 71–78%.** Standard depthwise convolutions (DW-Std, using implicit GEMM (Chetlur et al., 2014) via PyTorch (Paszke et al., 2019)/cuDNN) consistently degrade by 71–78% when ker-

*Table 1.* WBMM-C $14 \times 14$ speedup relative to DW-Std $5 \times 5$ baseline across different batch sizes and feature map resolutions. WBMM's advantage increases with batch size due to batch-independent weight construction.

| Batch | $14{\times}14$ | $28{\times}28$ | $56{\times}56$ | $112{\times}112$ | $224{\times}224$ |
|---|---|---|---|---|---|
| 4 | $0.45\times$ | $0.37\times$ | $1.41\times$ | $1.64\times$ | $2.02\times$ |
| 16 | $0.62\times$ | $1.56\times$ | $1.80\times$ | $2.00\times$ | $2.07\times$ |
| 64 | $1.86\times$ | $1.85\times$ | $2.01\times$ | $2.01\times$ | $2.03\times$ |
| 128 | $3.17\times$ | $2.00\times$ | $2.06\times$ | $1.97\times$ | $2.01\times$ |
| 256 | $\mathbf{3.72\times}$ | $\mathbf{2.02\times}$ | $\mathbf{2.02\times}$ | $\mathbf{1.95\times}$ | $-$ |

Note: "–" indicates OOM at batch=256 for DW-Std.

nel size increases from $5 \times 5$ to $13 \times 13$, across all tested batch sizes and feature map resolutions ($\geq 28 \times 28$). For $27 \times 27$ kernels, degradation reaches 92–94%. This degradation is *inherent* to the gather-based computation pattern and represents the baseline efficiency that any acceleration technique must improve upon.

**Finding 2: LKA Becomes Counterproductive on Large Feature Maps.** The Large Kernel Acceleration CUDA kernels (LKA), employed by RepLKNet and UniRepLKNet for training workloads, use a fixed tiling tuned for small feature maps that no longer matches the workload at large resolution, so the kernels fall behind the standard baseline. At $224 \times 224$ resolution and batch=128, LKA $7 \times 7$ is **80% slower** than baseline, and LKA $13 \times 13$ is **89% slower**. These kernels should be avoided for inference on large feature maps.

**Finding 3: WBMM Accelerates with Larger Windows.** Unlike depthwise convolutions that degrade with kernel size, WBMM *accelerates* with larger window sizes. Compared to the DW-Std $5 \times 5$ baseline, WBMM-C $14 \times 14$ (window) provides a $7.8\times$ larger *per-layer receptive field* (196 vs. 25 positions per layer) while achieving faster speed on feature maps $\geq 28 \times 28$ at batch $\geq 16$ (and $\geq 56 \times 56$ at all tested batch sizes; see Table 1).

**Finding 4: WBMM's Advantage Scales with Batch Size.** Table 1 shows that WBMM-C $14 \times 14$ (window) achieves increasing speedup over DW-Std $5 \times 5$ as batch size grows: from $0.45\times$ at batch=4 to $\mathbf{3.72\times}$ at batch=256 on $14 \times 14$ *feature maps* (note: window size and feature-map size coincide in this column), confirming WBMM's batch-independent weight construction advantage.

### 4.2. Ablation Studies

We establish UniRepLKNet-T as our baseline model and systematically examine critical design components.

*Table 2.* Ablation studies on feature extraction and window interaction. Of the three window-interaction variants, the serial $3 \times 3$ attains the best mIoU (with Top-1 comparable to the other two) but the highest cost; the configuration actually used in our models is selected from the across-block exploration in Table 3.

| Component | Method | Top-1 (%) | mIoU (%) |
|---|---|---|---|
| Feature Extract. | Full connection | 80.89±0.02 | 44.02±0.05 |
| | Relative pos. bias | 82.71±0.04 | 45.50±0.07 |
| | Rel. pos. + shortcut | 82.72±0.03 | 45.79±0.07 |
| Window Interact. | $3 \times 3$ dw parallel | 83.11±0.02 | 47.90±0.06 |
| | $3 \times 3$ dw serial | 83.21±0.03 | 48.01±0.03 |
| | Half ch. $3 \times 3$ dw | 83.22±0.01 | 46.21±0.04 |

#### 4.2.1. FEATURE EXTRACTION AND WINDOW INTERACTION

Table 2 presents our findings. For feature extraction, incorporating position-aware weights through relative position bias significantly improved performance (82.71±0.04% Top-1, 45.50±0.07% mIoU) compared to basic full connection (80.89±0.02% Top-1, 44.02±0.05% mIoU). The substantial performance gap (nearly 2 percentage points) highlights the importance of spatial relationship modeling.

For window interaction, we compare three ways of combining a $3 \times 3$ depthwise convolution with WBMM: a *parallel* branch, a *serial* arrangement in which the $3 \times 3$ convolution follows WBMM, and *channel splitting* in which half the channels take the $3 \times 3$ path. The serial arrangement attains the best mIoU (48.01±0.03%), ahead of parallel (47.90±0.06%) and well above channel splitting (46.21±0.04%). On Top-1, serial (83.21±0.03%) and channel splitting (83.22±0.01%) are essentially tied and both edge out parallel (83.11±0.02%). Since channel splitting falls clearly behind on dense prediction while serial leads parallel on both metrics, the serial integration of local and global features provides more effective information flow than fusing them in parallel. Applying a $3 \times 3$ after *every* WBMM, however, increases both training and inference time, so we do *not* adopt this per-block serial design; instead, the mixed block patterns in Table 3 distribute WBMM and $3 \times 3$ across stages, which—tuned per task—match the per-block serial design's classification accuracy (the 0.01% Top-1 gap is negligible) while *surpassing* its mIoU (48.32 vs. 48.01), all at lower training and inference cost.

#### 4.2.2. OPTIMAL ARCHITECTURE DESIGN

Table 3 explores these mixed block patterns across stages. For image classification, performance peaked (83.21% Top-1) with configuration $[D,D,D]_{S1} \rightarrow [W,D,W]_{S2} \rightarrow [W \leftrightarrow D]_{S3} \rightarrow [W,W,W]_{S4}$. For semantic segmentation, optimal performance (48.32% mIoU) was achieved with $[W,D,W]_{S1} \rightarrow [W,D,W]_{S2} \rightarrow [W \leftrightarrow D]_{S3} \rightarrow [W,W,W]_{S4}$.

*Table 3.* Architecture exploration. "W" denotes WBMM block ($7 \times 7$ window), "D" denotes $3 \times 3$ depthwise convolution. Highlighted rows show optimal configurations.

| Block Pattern by Stage | | | | WBMM | |
|---|---|---|---|---|---|
| S1 | S2 | S3 | S4 | Top-1 | mIoU |
| W,W,W | W,W,W | W↔W | W,W,W | 82.72 | 45.81 |
| W,W,W | W,W,W | W↔D | W,W,W | 82.99 | 47.18 |
| D,D,D | W,W,W | W↔D | W,W,W | 83.18 | 47.59 |
| D,D,D | W,D,W | W↔D | W,W,W | **83.21** | 47.63 |
| W,D,W | W,W,W | W↔D | W,W,W | 83.02 | 47.52 |
| W,D,W | W,D,W | W↔D | W,W,W | 83.02 | **48.32** |
| W,D,W | W,D,W | W↔D | W,D,W | 83.11 | 47.89 |
| D,W,D | W,D,W | W↔D | W,W,W | 82.98 | 48.19 |

Note: S1–S4 represent network stages at different resolutions. "↔" indicates alternating operations.

*Table 4.* Ablation on hierarchical window reparameterization on ADE20K with WBMM-T. "G+L" denotes hierarchical reparameterization combining global $14 \times 14$ windows with local $7 \times 7$ sub-windows.

| Configuration | S1 | S2 | S3 | S4 | mIoU (%) |
|---|---|---|---|---|---|
| Pure $7 \times 7$ (baseline) | 7 | 7 | 7 | 7 | 48.0±0.03 |
| Pure $14 \times 14$ | 14 | 14 | 14 | 14 | 47.9±0.05 |
| Pure $14 \times 14$ (S4: 7) | 14 | 14 | 14 | 7 | 47.8±0.04 |
| Optimal Config | G+L | G+L | 14 | 7 | **48.8±0.06** |
| All Hierarchical | G+L | G+L | G+L | 7 | 48.1±0.05 |

This task-dependent divergence reflects different downstream use of S1 features: classification uses only the globally pooled deepest feature (favoring local $3 \times 3$ extraction at S1), whereas FPN/UPerNet-style decoders fuse S1 directly with upsampled deeper features, benefiting from the broader spatial context WBMM provides at the highest-resolution stage.

#### 4.2.3. HIERARCHICAL WINDOW REPARAMETERIZATION

Table 4 shows that applying hierarchical reparameterization (G+L) selectively to S1–S2 stages achieves 48.8±0.06% mIoU, providing **0.8%** improvement over the $7 \times 7$ baseline and 0.9% over pure $14 \times 14$ configuration.

### 4.3. ImageNet Classification

Table 5 compares WBMM and UniRepLKNet on ImageNet-1K. WBMM matches or exceeds accuracy while offering significant efficiency gains: training speedup of **1.31–1.88×**, memory reduction of 8–12%, and comparable accuracy (within 0.1%).

### 4.4. Semantic Segmentation on ADE20K

We evaluate WBMM on ADE20K semantic segmentation using UPerNet (Xiao et al., 2018) with both Tiny and Small model variants. Semantic segmentation processes high-

*Table 5.* Comparison on ImageNet-1K classification. WBMM achieves 1.31–1.88× training speedup with comparable accuracy. Training: 8× A800 GPUs. LKA: Large Kernel Acceleration. Base-scale results (WBMM-B) are reported in Section F.

| Model | Params (M) | FLOPs (G) | Mem. (GB) | Time (m:s) | Acc. (%) | Speedup |
|---|---|---|---|---|---|---|
| WBMM-P | 10.6 | 1.6 | 8.61 | 3:46 | 80.3 | Baseline |
| UniRepLKNet-P (LKA) | 10.7 | 1.6 | 9.45 | 4:56 | 80.2 | 1.31× |
| UniRepLKNet-P (no LKA) | 10.7 | 1.6 | 9.45 | 5:53 | 80.2 | 1.56× |
| WBMM-N | 18.1 | 2.7 | 10.01 | 4:12 | 81.7 | Baseline |
| UniRepLKNet-N (LKA) | 18.3 | 2.8 | 11.33 | 6:17 | 81.6 | 1.50× |
| UniRepLKNet-N (no LKA) | 18.3 | 2.8 | 11.33 | 7:54 | 81.6 | 1.88× |
| WBMM-T | 31.0 | 4.8 | 15.04 | 6:18 | 83.2 | Baseline |
| UniRepLKNet-T (LKA) | 31.0 | 4.9 | 16.58 | 9:03 | 83.2 | 1.44× |
| UniRepLKNet-T (no LKA) | 31.0 | 4.9 | 16.58 | 11:03 | 83.2 | 1.75× |
| WBMM-S | 55.6 | 9.0 | 20.43 | 8:35 | 83.9 | Baseline |
| UniRepLKNet-S (LKA) | 55.6 | 9.1 | 22.27 | 12:10 | 83.9 | 1.42× |
| UniRepLKNet-S (no LKA) | 55.6 | 9.1 | 22.27 | 14:16 | 83.9 | 1.66× |

*Table 6.* Semantic segmentation on ADE20K validation set with UPerNet. Both Tiny (T) and Small (S) variants are evaluated. WBMM with hierarchical reparameterization achieves **higher mIoU** than UniRepLKNet.

| Method | Params (M) | | FLOPs (G) | | mIoU SS | | mIoU MS | |
|---|---|---|---|---|---|---|---|---|
| | T | S | T | S | T | S | T | S |
| UniRepLKNet | 62 | 87 | 946 | 1036 | 48.6 | 50.5 | 49.1 | 51.0 |
| WBMM (7 × 7) | 62 | 87 | 944 | 1033 | 48.3 | 50.2 | 48.8 | 50.5 |
| WBMM (Hier) | 66 | 92 | 948 | 1038 | **48.8** | **50.6** | **49.3** | **51.2** |

Note: "Hier" denotes hierarchical reparameterization combining 14 × 14 windows with 7 × 7 sub-windows.

resolution feature maps with variable-sized inputs, which WBMM handles seamlessly via its zero-padding mechanism (Section 3.4).

Table 6 shows that WBMM with hierarchical reparameterization achieves higher mIoU than UniRepLKNet on both Tiny and Small variants. The pure 7 × 7 variant matches FLOPs and parameters of UniRepLKNet within 0.3 mIoU, and adding hierarchical reparameterization closes this gap without slowing inference.

### 4.4.1. INFERENCE SPEED ANALYSIS

We provide comprehensive throughput measurements for both Tiny and Small models across multiple input resolutions (512×512, 512×1024, 1024×1024) and batch sizes (2, 4, 8, 16) in Section J, demonstrating three key findings: (1) WBMM with 7 × 7 windows approaches the theoretical upper bound of Conv 3 × 3 speed while enabling dense intra-window interactions; (2) Both WBMM configurations—7 × 7 windows and hierarchical reparameterization with 14×14 windows—deliver consistent speedup over UniRepLKNet across all tested resolutions and batch sizes; (3) WBMM excels particularly at high resolution (e.g., 1024 × 1024) where LKA-accelerated UniRepLKNet suffers from severe performance degradation.

*Table 7.* Object detection on COCO with Cascade Mask R-CNN. WBMM with hierarchical reparameterization achieves **higher AP** than UniRepLKNet.

| Method | Params (M) | | FLOPs (G) | | $AP^{box}$ | | $AP^{mask}$ | |
|---|---|---|---|---|---|---|---|---|
| | T | S | T | S | T | S | T | S |
| UniRepLKNet | 89 | 113 | 749 | 835 | 51.8 | 53.0 | 44.9 | 45.9 |
| WBMM (7 × 7) | 89 | 113 | 747 | 833 | 51.6 | 52.8 | 44.8 | 45.6 |
| WBMM (Hier) | 92 | 118 | 751 | 837 | **51.9** | **53.1** | **45.1** | **46.1** |

Note: FLOPs computed at 1280 × 800 resolution.

### 4.5. Object Detection on COCO

We further evaluate WBMM on COCO object detection with Cascade Mask R-CNN (Cai & Vasconcelos, 2019), using the same dense-prediction architecture as segmentation. WBMM with hierarchical reparameterization achieves **higher AP** than UniRepLKNet: 51.9%/45.1% (box/mask) for Tiny and 53.1%/46.1% for Small (Table 7), while maintaining competitive inference speed.

### 4.6. Hardware Generalization

We evaluate WBMM on diverse hardware platforms; detailed benchmarks are in Section K. On a single **NVIDIA A800 GPU** (FP32, batch=128), WBMM achieves 1.01–1.28× over LKA-accelerated UniRepLKNet and 1.23–1.41× over the non-accelerated version. On **Intel Core i7-13700K CPU** (FP32), where specialized large kernel acceleration is unavailable, WBMM achieves 1.03–1.48× via highly optimized BLAS libraries. On **NVIDIA Jetson Orin Nano** (8GB), WBMM achieves 1.19–3.12× on GPU (FP16) and 1.33–2.44× on CPU (INT8), confirming compatibility with quantization.

## 5. Conclusion

We presented Windowed Batch Matrix Multiplication (WBMM), which accelerates large-kernel convolutions by traversing a compact parameter table instead of gathering scattered input neighborhoods, turning a memory-bound gather into a compute-bound batched matrix multiplication. WBMM thus speeds up with larger windows and with batch size (up to 3.72× at batch=256), and—unlike LKA-accelerated convolutions that slow down on large feature maps—maintains consistent efficiency while providing a 7.8× larger per-layer receptive field than DW-Std 5 × 5. Combined with inter-block communication, inference caching, and hierarchical reparameterization, WBMM matches UniRepLKNet on ImageNet-1K and outperforms it on ADE20K and COCO with a 1.31–1.88× training speedup, while **generalizing across GPU, CPU, and edge devices** without specialized kernels. We discuss limitations and future extensions in Section O.

## Acknowledgements

This work was supported in part by the Anhui Province Major Science and Technology Project under Grant JZ2024AKKZ0025. The numerical computations were carried out on the HPC Platform of Hefei University of Technology. The authors thank Professor Wengang Zhou of the University of Science and Technology of China for his valuable guidance on the writing of this manuscript, and the anonymous ICML 2026 reviewers for their constructive comments that substantially improved the paper.

## Impact Statement

This paper presents work advancing the field of Machine Learning. The efficiency gains from WBMM could reduce energy consumption and carbon footprint in training and deploying vision models, potentially democratizing access to large-receptive-field models for researchers with limited computational resources.

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

## A. Appendix Overview

This appendix provides comprehensive supplementary materials organized as follows:

- **Section B**: Fair comparison protocol ensuring reproducibility.

- **Section C**: Controlled cross-operator comparison (SLaK, DW variants) under identical architecture.

- **Section D**: WBMM vs. non-overlapping window attention under matched parameters.

- **Section E**: ConvNeXt-T drop-in replacement experiment.

- **Section F**: Base-scale (~100M) ImageNet-1K results.

- **Section G**: Padding-strategy ablation on ADE20K.

- **Section H**: Analysis of learned weight matrices ($M$).

- **Section I**: Complete operator-level benchmarks across all batch sizes.

- **Section J**: Detailed segmentation inference speed analysis.

- **Section K**: Hardware generalization benchmarks on GPU, CPU, and edge devices.

- **Section L**: Complete training configurations.

- **Section M**: Model architecture specifications.

- **Section N**: Hierarchical reparameterization algorithm details.

- **Section O**: Limitations and future work.

## B. Fair Comparison Protocol

To ensure fair evaluation, we maintain identical framework configurations: same backbone structure, training protocols, hyperparameters, data preprocessing, and optimization strategies. The modifications include: (1) replacing large depthwise convolution kernels with WBMM operations, and (2) adapting the mixing strategy of how WBMM blocks and $3 \times 3$ depthwise convolution blocks are arranged across different layers. The implementation follows the UniRepLKNet codebase.

**Note on Structural Reparameterization.** UniRepLKNet employs extensive structural reparameterization with multiple parallel branches during training, adding significant computational overhead. WBMM uses minimal reparameterization (only applied to S4 stage for Pico/Nano variants in classification tasks). Therefore, reported training speedups represent conservative estimates.

## C. Controlled Cross-Operator Comparison

To isolate WBMM's contribution from architecture-level design choices, we conduct a strictly controlled experiment in which all configurations share identical architecture, training protocol, hyperparameters, data preprocessing, and optimization; *only the spatial mixing operator inside the spatial blocks differs*. To enable a single comparison that is meaningful for both classification and dense prediction, all entries in Table 8 use the dense-prediction architecture variant (Tiny* in Table 20: [W,D,W] at S1, [W,D,W] at S2, [W↔D]×9 at S3, [W,W,W] at S4), in which extra spatial-mixing blocks are placed at the high-resolution stage S1 to favor segmentation/detection. This is why the WBMM-T ($w$=7) Top-1 in Table 8 (83.0) is slightly lower than the value reported in Table 5 of the main text (83.2): the classification-tuned architecture used there (Tiny in Table 20: [D,D,D] at S1) replaces the S1 WBMM blocks with pure $3 \times 3$ depthwise convolutions, which is empirically better for ImageNet classification (see also Table 3). The same architecture is applied to every operator variant in Table 8, so the comparison among operators remains apples-to-apples. Training uses $8\times$ A800 GPUs; inference speed is measured on a single A800 (FP32, batch=128). "Hier" denotes hierarchical window reparameterization (see Section 3.6) with $14 \times 14$ global windows and $7 \times 7$ local sub-windows fused at inference into a single matrix without slowing inference. "LKA" denotes the Large Kernel Acceleration CUDA kernels of RepLKNet/UniRepLKNet (used consistently throughout

this paper); "reparam" denotes the UniRepLKNet structural reparameterization with `kernel_sizes=[5,7,3,3,3]` and `dilates=[1,2,3,4,5]`. "SLaK-style" uses parallel rectangular decomposition ($5 \times k + k \times 5 + 5 \times 5$ with $k \in \{51, 49, 47, 13\}$).

*Table 8.* Controlled cross-operator comparison on the Tiny model. All entries share identical architecture, training, and hyperparameters; only the spatial operator differs. "Time" is per-epoch ImageNet training time (min:sec); "Spd" is inference throughput on a single A800.

| Operator | Params (M) | FLOPs (G) | Mem (GB) | Time | Top-1 (%) | mIoU (%) | Spd (img/s) |
|---|---|---|---|---|---|---|---|
| WBMM-T ($w$=7) | 31.0 | 4.8 | 15.16 | 6:20 | 83.0 | 48.3 | 1833.1 |
| WBMM-T ($w$=14, Hier) | 33.0 | 5.1 | 15.87 | 6:31 | 83.2 | **48.8** | **1842.2** |
| $13 \times 13$ DW + LKA | 31.0 | 5.0 | 14.77 | 6:43 | 83.1 | 48.3 | 1661.6 |
| $13 \times 13$ DW (no LKA) | 31.0 | 5.0 | 14.77 | 9:11 | 83.1 | 48.3 | 1305.3 |
| $13 \times 13$ DW + reparam + LKA | 31.6 | 5.1 | 17.23 | 10:21 | 83.2 | 48.7 | 1661.6 |
| $13 \times 13$ DW + reparam (no LKA) | 31.6 | 5.1 | 17.23 | 12:46 | 83.2 | 48.7 | 1305.3 |
| SLaK-style (in WBMM design) | 32.1 | 5.4 | 16.19 | 15:27 | 83.2 | 47.2 | 772.1 |
| SLaK-Original (all stages) | 33.7 | 5.9 | 16.91 | 22:03 | 83.3 | 47.4 | 549.0 |
| Win7 Transformer (MLP $r$=2.5) | 31.2 | 5.1 | 16.65 | 8:38 | 83.3 | 48.1 | 1245.1 |

Note: All entries here use the dense-prediction architecture (Tiny$^*$ in Table 20: [W,D,W] at S1) for an apples-to-apples comparison between operators on *both* classification and segmentation. The classification-tuned architecture (Tiny in Table 20: [D,D,D] at S1) is used in Table 5 of the main text and yields the higher Top-1 (83.2) for WBMM-T ($w$=7). The slight Time/Mem difference vs. Table 5 (6:20/15.16 vs. 6:18/15.04) reflects this architectural difference and is consistent within run-to-run variation.

**Key observations.** (1) *WBMM $w$=7 vs. $13 \times 13$ DW.* Comparable classification (83.0 vs. 83.1) and identical mIoU (48.3), but WBMM is faster in training (6:20 vs. 6:43 with LKA / 9:11 without) and inference (1833 vs. 1662/1305 img/s) with lower memory. (2) *WBMM $w$=14 Hier vs. $13 \times 13$ DW + reparam.* UniRepLKNet's complex reparameterization recovers 48.7 mIoU at a severe training cost (10:21–12:46 vs. 6:31) and +1.36 GB memory; WBMM Hier still leads +0.1 mIoU at much lower overhead. (3) *WBMM vs. SLaK variants.* Despite an effective receptive field of 51, both SLaK variants underperform on dense prediction (47.2/47.4 vs. 48.3/48.8 mIoU) and are 2.4–3.4× slower in both training and inference. (4) *WBMM vs. window attention.* See Section D.

## D. WBMM vs. Non-Overlapping Window Attention

To compare input-independent (WBMM) and input-dependent (attention) weight construction *at the operator level*, we replace only the WBMM blocks with non-overlapping $7 \times 7$ window self-attention blocks. The MLP expansion ratio of the attention variant is set to 2.5 so that the parameter count and FLOPs match WBMM. Everything else—overall architecture, the inter-block $3 \times 3$ depthwise mixing, training schedule and hyperparameters—is held identical.

The relevant rows from Table 8: Win7 Transformer (MLP $r$=2.5) reaches 31.2 M params, 5.1 G FLOPs, Top-1 83.3, mIoU 48.1, with per-epoch time 8:38 and inference speed 1245.1 img/s. WBMM-T at matched scale reaches Top-1 83.0, mIoU 48.3 ($w$=7) / 48.8 ($w$=14, Hier), with training 6:20/6:31 and inference 1833.1/1842.2 img/s. Window attention obtains a small classification gain (+0.3 Top-1 under the dense-prediction architecture used in Table 8; see the note under that table), but falls behind on dense prediction ($-0.2$ to $-0.7$ mIoU). WBMM is **1.32–1.36×** faster in training and **1.47–1.48×** faster in inference, with ~0.8–1.5 GB lower memory.

**Where the gap comes from.** Window attention computes query–key products for every sample and every window at every forward pass ($O(B \cdot N_h N_w \cdot d^2)$, with $N_h N_w$ windows per sample), and its softmax couples positions in a way that requires online normalization when tiled. WBMM's weights are input-independent and constructed once at $O(C \cdot d^2)$, shared across all batches and windows; cached at inference, the cost is amortized to zero. The empirical gap is consistent with this asymptotic difference, and grows with batch size. Table 9 summarizes these property differences.

*Table 9.* Property comparison between WBMM and window attention.

| Property | WBMM | Window Attention |
|---|---|---|
| Weight construction | $O(C \cdot d^2)$, batch-independent | $O(B \cdot N_h N_w \cdot d^2)$ |
| Softmax | None | Required |
| Batch scalability | Cost-amortized ($M$ shared) | Linear in $B \cdot N_h N_w$ |
| Tiling | Simple (pure linear) | Complex (online softmax) |
| Adaptive to input | No (fixed at inference) | Yes |

**When input-dependence may help.** Content-adaptive routing tasks (e.g., VQA, referring expression) and tasks with extreme intra-image variation (e.g., satellite imagery) may benefit from input-dependent weights. A lightweight input-dependent gate over WBMM's cached $M$ would offer a hybrid design and is a natural extension.

## E. Cross-Architecture Validation on ConvNeXt-T

To verify that WBMM's efficiency is not specific to the UniRepLKNet backbone, we perform a drop-in replacement on ConvNeXt-T: all $7 \times 7$ depthwise convolutions are replaced with WBMM blocks ($w=7$) interleaved with $3 \times 3$ depthwise mixing blocks. No other change is made to the ConvNeXt training recipe.

*Table 10.* Cross-architecture drop-in replacement on ConvNeXt-T. "Time" is per-epoch ImageNet training time (min:sec).

| Model | Params (M) | FLOPs (G) | Mem (GB) | Top-1 (%) | mIoU (%) | Time |
|---|---|---|---|---|---|---|
| ConvNeXt-T ($7 \times 7$ DW) | 28.6 | 4.5 | 17.93 | 82.1 | 46.0 | 8:30 |
| ConvNeXt-T $\rightarrow$ WBMM ($w=7$) + $3\times3$ mix | 29.1 | 4.4 | 18.20 | 82.1 | **46.2** | **5:08** |

As shown in Table 10, classification accuracy is preserved exactly (82.1 Top-1), segmentation is slightly higher (46.2 vs. 46.0 mIoU), and training is $1.66\times$ **faster** (5:08 vs. 8:30). This confirms that WBMM's efficiency gains are architecture-agnostic and require no task-specific tuning.

## F. Base-Scale Results

We extend the comparison to the Base scale ($\sim$100 M parameters) on ImageNet-1K (300 epochs, identical hyperparameters), with results summarized in Table 11:

*Table 11.* Base-scale ImageNet-1K results.

| Model | Params (M) | FLOPs (G) | Mem (GB) | Time | Top-1 (%) |
|---|---|---|---|---|---|
| WBMM-B ($w=7$) | 97.9 | 15.9 | 26.38 | **11:24** | **83.9** |
| UniRepLKNet-B | 98.0 | 16.1 | 29.01 | 15:46 | 83.8 |

WBMM-B achieves $1.38\times$ **training speedup** (11:24 vs. 15:46 per epoch) with **9.1% lower memory** (26.38 vs. 29.01 GB) and $+0.1\%$ higher accuracy (83.9 vs. 83.8). Notably, UniRepLKNet-B does not improve over its Small variant (83.8 vs. 83.9) on ImageNet-1K alone, suggesting saturation at this data scale—the original paper uses ImageNet-22K pre-training for Base models. WBMM-B maintains Small-level accuracy on ImageNet-1K alone, showing slightly better scaling. ImageNet-22K pre-training experiments are planned future work.

## G. Padding-Strategy Ablation

WBMM relies on padding to align spatial dimensions with the window size. On ADE20K with UPerNet (WBMM-T Hier, $w=14$, variable input sizes), we compare three padding strategies. Zero-padding is empirically the best of the three strategies tested. Several factors contribute to this result: (1) four-stage downsampling ($4\times$ to $32\times$) confines the padded region to $< 0.1\%$ of the S4 feature map; (2) UPerNet's multi-scale fusion dilutes any boundary artifacts; (3) the inter-block $3 \times 3$ depthwise mixing smooths boundary discontinuities; (4) reflection and replication padding introduce correlated values that interfere with the position-aware weights in $R$, whereas zeros provide a clean neutral signal. Combined with hierarchical reparameterization, which reinforces local patterns, boundary artifacts are kept negligible. The quantitative results are summarized in Table 12.

*Table 12.* Padding-strategy ablation on ADE20K (single-scale mIoU).

| Padding strategy | mIoU (SS, %) |
|---|---|
| Zero-padding (ours) | **48.8** |
| Reflection padding | 48.6 |
| Replication padding | 48.4 |

The gap between zero-padding and the alternatives is small (0.2–0.4 mIoU) but consistent across three independent runs, and we therefore use zero-padding throughout the paper. This comparison is performed with the hierarchical $14 \times 14 + 7 \times 7$ configuration at the high-resolution stages, where the padding fraction is highest; for smaller windows the absolute differences shrink further because less padding is required.

## H. Analysis of Learned Weight Matrices

To understand what WBMM learns, we visualize the effective weight matrix $M$ at inference and observe three interpretable patterns; Figure 4 summarizes the visualization.

**(a) Locality.** Across stages, $M$ exhibits strong diagonal dominance with diagonal-to-off-diagonal ratios of $41$–$58\times$. The mean off-diagonal magnitude $|M(p, q)|$ decays by over $90\%$ within Euclidean distance 2, confirming that WBMM *automatically* learns a locality prior despite having no explicit regularization toward locality.

**(b) Channel specialization.** Different channels exhibit clearly distinct preferences—horizontal, vertical, and diagonal—that resemble oriented edge detectors. This diversity arises naturally from the per-channel parameterization of $R$, in contrast to attention which shares weights across channels within a head.

**(c) Frequency selectivity.** Channels split into low-pass and high-pass families (e.g., low-frequency energy ratio LF= $83\%$ vs. LF= $9\%$). Shallow stages (S2) bias toward high-pass, while deeper stages (S3/S4) are more balanced. The $(2w - 1)^2 = 169$ parameters per channel are clearly sufficient to encode this diversity.

These patterns emerge *without* explicit regularization, suggesting WBMM provides an efficient and expressive basis for learning spatial operators.

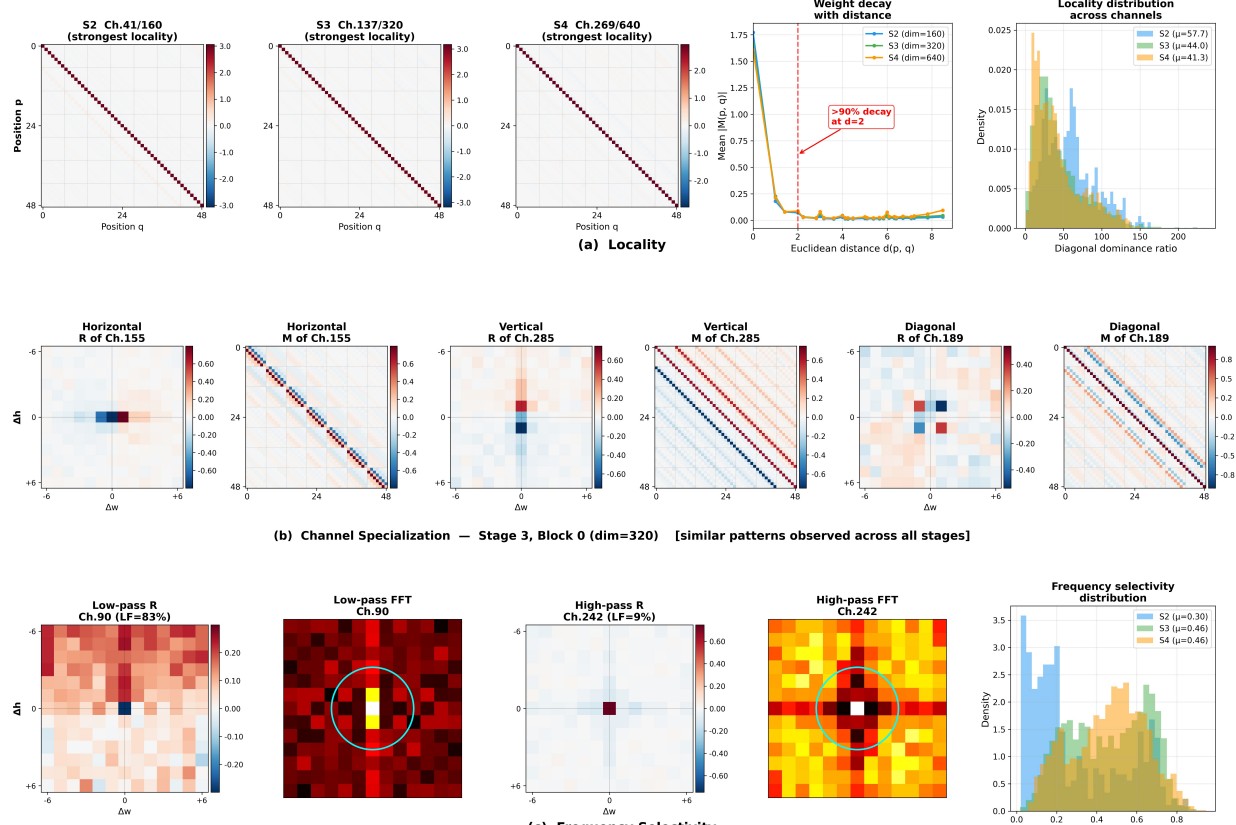

*Figure 4.* **Interpretable structure of learned WBMM weight matrices $M$.** (a) Locality: $M$ exhibits strong diagonal dominance and $> 90\%$ weight decay within distance 2. (b) Channel specialization: channels learn distinct horizontal, vertical, and diagonal patterns resembling oriented edge detectors. (c) Frequency selectivity: low-pass vs. high-pass channels coexist, with shallow stages biased toward high-pass and deeper stages more balanced.

# I. Complete Operator-Level Benchmark Across All Batch Sizes

We present comprehensive operator-level benchmark results across all tested batch sizes (4, 16, 64, 128, 256). All measurements use 256 channels, FP32 precision on a single NVIDIA A800 GPU (80GB), with 5 independent runs and IQR-based outlier removal. These results provide the complete data underlying Figure 3 and Table 1 in the main text.

**Why Single-Layer Operator Benchmarks.** We conduct operator-level benchmarks on single-layer feature maps for several important reasons:

1. **Isolation of computational properties**: Single-layer testing isolates the fundamental computational characteristics of WBMM vs depthwise convolutions, free from confounding factors like network depth, skip connections, or layer interactions.

2. **Architecture-agnostic conclusions**: Results from isolated operator benchmarks generalize across different CNN architectures (ConvNeXt, ResNet, etc.), not just UniRepLKNet.

3. **Controlled experimental conditions**: Fixed channel count (256) and systematic variation of batch size and feature map resolution enable precise characterization of scaling behavior.

4. **Reproducibility**: Isolated benchmarks are easier to reproduce and verify independently.

## I.1. Benchmark Analysis Summary

The comprehensive benchmark results in Table 13 confirm the following architecture-agnostic properties:

**Key Finding 1: Standard Convolution Degradation is Inherent.** The 71–78% throughput degradation of DW-Std from $5 \times 5$ to $13 \times 13$ (and 92–94% for $27 \times 27$) holds across all tested batch sizes and feature map resolutions $\geq 28 \times 28$, confirming that this penalty is inherent to the gather-based computation pattern (cf. Section 4.1).

**Key Finding 2: LKA Has a Crossover Point.** LKA shows effectiveness only on small feature maps. At batch$\geq$64, LKA provides speedup only for feature maps $\leq 14 \times 14$. On large feature maps ($\geq 112 \times 112$), LKA is 63–99% slower than baseline.

**Key Finding 3: WBMM Scaling.** WBMM-C $14 \times 14$ achieves $+56\%-+108\%$ speedup over baseline on feature maps $\geq 28 \times 28$ at batch$\geq$16, peaking at $+272\%$ on $14 \times 14$ maps at batch=256.

**Key Finding 4: Caching Benefit.** WBMM-C provides up to $\sim 3\times$ speedup over WBMM-NC on the smallest feature maps (e.g., a single $14 \times 14$ window at small batch), where weight construction overhead is proportionally larger.

# J. Segmentation Inference Speed Analysis

We provide comprehensive throughput measurements for both Tiny and Small model variants on ADE20K semantic segmentation. All experiments are conducted on a **single NVIDIA A800 GPU (80GB)** with FP32 precision. These results support the claims made in Section 4.4.

## J.1. Tiny Model Results

Table 14 reports the throughput measurements for the Tiny models.

**Key Observations for Tiny Model.** **(1) WBMM approaches theoretical upper bound**: At $512 \times 512$ with batch=16, WBMM-T with $7 \times 7$ windows achieves 104.6 FPS, only 2.0% slower than the Conv $3 \times 3$ upper bound (106.7 FPS) despite providing a $5.4\times$ larger *per-layer receptive field* (49 vs. 9 positions per layer). This demonstrates that WBMM's regular memory access pattern nearly eliminates the performance penalty traditionally associated with large receptive fields.

**(2) WBMM excels at high resolution**: At $1024 \times 1024$ resolution—critical for dense prediction quality—hierarchical WBMM achieves 30.4 FPS (batch=16), **outperforming** even Conv $5 \times 5$ (29.8 FPS) while providing a $7.8\times$ larger per-layer receptive field (196 vs. 25 positions per layer). This validates WBMM's core design principle: efficiency improves rather than degrades with larger feature maps.

Table 13. **Unified operator-level benchmark**: throughput (kFPS) across all batch sizes, methods, and feature map resolutions. All measurements use 256 channels, FP32 precision on single NVIDIA A800 GPU (80GB). Δ: speedup relative to DW-Std $5 \times 5$ baseline. "–" indicates out-of-memory or invalid configuration.

| Batch | Method | Size | 7×7 | | 14×14 | | 28×28 | | 56×56 | | 112×112 | | 224×224 | |
|---|---|---|---|---|---|---|---|---|---|---|---|---|---|---|
| | | | kFPS | Δ | kFPS | Δ | kFPS | Δ | kFPS | Δ | kFPS | Δ | kFPS | Δ |
| B=4 | DW-Std | 3 × 3 | 140 | –7% | 140 | 0% | 145 | +85% | 65 | +186% | 17 | +124% | 5.3 | +180% |
| | DW-Std | 5 × 5 | 150 | base | 140 | base | 78 | base | 23 | base | 7.4 | base | 1.9 | base |
| | DW-Std | 7 × 7 | 119 | –21% | 99 | –30% | 48 | –39% | 14 | –38% | 4.5 | –39% | 1.14 | –40% |
| | DW-Std | 13 × 13 | 106 | –29% | 59 | –58% | 19 | –76% | 6.4 | –72% | 1.7 | –77% | 0.42 | –78% |
| | DW-Std | 27 × 27 | 46 | –69% | 19 | –86% | 6.2 | –92% | 1.7 | –93% | 0.43 | –94% | 0.11 | –94% |
| | DW-LKA | 7 × 7 | 78 | –48% | 29 | –79% | 7.5 | –90% | 1.2 | –95% | 0.18 | –98% | 0.02 | –99% |
| | DW-LKA | 13 × 13 | 77 | –49% | 25 | –82% | 5.3 | –93% | 0.78 | –97% | 0.10 | –99% | 0.014 | –99% |
| | DW-LKA | 27 × 27 | 78 | –48% | 24 | –83% | 3.4 | –96% | 0.44 | –98% | 0.054 | –99% | – | – |
| | WBMM-C | 7 × 7 | 95 | –37% | 55 | –61% | 42 | –47% | 27 | +18% | 8.4 | +14% | 2.6 | +38% |
| | WBMM-C | 14 × 14 | – | – | 63 | –55% | 29 | –63% | **32** | **+40%** | **12** | **+64%** | **3.8** | **+101%** |
| | WBMM-NC | 7 × 7 | 66 | –56% | 47 | –66% | 35 | –55% | 24 | +4% | 8.9 | +20% | 2.6 | +37% |
| | WBMM-NC | 14 × 14 | – | – | 21 | –85% | 15 | –81% | 16 | –31% | 9.9 | +34% | 3.5 | +84% |
| B=16 | DW-Std | 3 × 3 | 558 | +22% | 579 | +126% | 256 | +182% | 79 | +168% | 21 | +184% | 5.4 | +185% |
| | DW-Std | 5 × 5 | 460 | base | 256 | base | 91 | base | 29 | base | 7.6 | base | 1.9 | base |
| | DW-Std | 7 × 7 | 352 | –24% | 184 | –28% | 55 | –39% | 18 | –39% | 4.6 | –40% | 1.14 | –40% |
| | DW-Std | 13 × 13 | 279 | –39% | 90 | –65% | 26 | –71% | 6.7 | –77% | 1.7 | –78% | 0.42 | –78% |
| | DW-Std | 27 × 27 | 84 | –82% | 26 | –90% | 6.8 | –93% | 1.7 | –94% | 0.43 | –94% | 0.11 | –94% |
| | DW-LKA | 7 × 7 | 313 | –32% | 117 | –54% | 30 | –67% | 4.9 | –83% | 0.72 | –91% | 0.10 | –95% |
| | DW-LKA | 13 × 13 | 313 | –32% | 101 | –60% | 21 | –77% | 3.1 | –89% | 0.42 | –94% | 0.054 | –97% |
| | DW-LKA | 27 × 27 | 313 | –32% | 94 | –63% | 14 | –85% | 1.8 | –94% | 0.22 | –97% | – | – |
| | WBMM-C | 7 × 7 | 401 | –13% | 191 | –25% | 126 | +39% | 39 | +31% | 10 | +36% | 2.6 | +40% |
| | WBMM-C | 14 × 14 | – | – | 158 | –38% | **142** | **+56%** | **53** | **+80%** | **15** | **+99%** | **3.9** | **+108%** |
| | WBMM-NC | 7 × 7 | 279 | –39% | 158 | –38% | 110 | +21% | 37 | +26% | 10 | +35% | 2.7 | +40% |
| | WBMM-NC | 14 × 14 | – | – | 77 | –70% | 72 | –21% | 39 | +32% | 14 | +82% | 3.9 | +103% |
| B=64 | DW-Std | 3 × 3 | 2315 | +85% | 1042 | +138% | 317 | +168% | 85 | +182% | 22 | +185% | 5.5 | +187% |
| | DW-Std | 5 × 5 | 1250 | base | 437 | base | 118 | base | 30 | base | 7.6 | base | 1.9 | base |
| | DW-Std | 7 × 7 | 895 | –28% | 279 | –36% | 72 | –39% | 18 | –39% | 4.6 | –40% | 1.14 | –40% |
| | DW-Std | 13 × 13 | 357 | –71% | 104 | –76% | 27 | –77% | 6.8 | –77% | 1.7 | –78% | 0.42 | –78% |
| | DW-Std | 27 × 27 | 110 | –91% | 28 | –94% | 7.0 | –94% | 1.7 | –94% | 0.43 | –94% | – | – |
| | DW-LKA | 7 × 7 | 1250 | 0% | 463 | +6% | 118 | 0% | 20 | –36% | 2.8 | –63% | 0.39 | –80% |
| | DW-LKA | 13 × 13 | 1225 | –2% | 403 | –8% | 83 | –30% | 12 | –59% | 1.7 | –78% | 0.22 | –89% |
| | DW-LKA | 27 × 27 | 1225 | –2% | 450 | +3% | 54 | –54% | 7.0 | –77% | 0.87 | –89% | – | – |
| | WBMM-C | 7 × 7 | 1008 | –19% | 508 | +16% | 157 | +33% | 42 | +39% | 11 | +38% | 2.7 | +40% |
| | WBMM-C | 14 × 14 | – | – | **812** | **+86%** | **219** | **+86%** | **61** | **+102%** | **15** | **+101%** | **3.9** | **+101%** |
| | WBMM-NC | 7 × 7 | 812 | –35% | 443 | +1% | 151 | +28% | 41 | +38% | 10 | +37% | 2.7 | +40% |
| | WBMM-NC | 14 × 14 | – | – | 361 | –17% | 159 | +35% | 56 | +84% | 15 | +96% | 3.8 | +101% |
| B=128 | DW-Std | 3 × 3 | 3289 | +111% | 992 | +147% | 333 | +176% | 87 | +184% | 21.8 | +185% | 5.46 | +187% |
| | DW-Std | 5 × 5 | 1563 | base | 402 | base | 121 | base | 31 | base | 7.6 | base | 1.91 | base |
| | DW-Std | 7 × 7 | 833 | –47% | 279 | –31% | 73 | –40% | 18 | –40% | 4.6 | –40% | 1.14 | –40% |
| | DW-Std | 13 × 13 | 330 | –79% | 107 | –73% | 27 | –78% | 6.8 | –78% | 1.7 | –78% | 0.42 | –78% |
| | DW-Std | 27 × 27 | 119 | –92% | 29 | –93% | 7.1 | –94% | 1.7 | –94% | 0.43 | –94% | – | – |
| | DW-LKA | 7 × 7 | 2119 | +36% | 530 | +32% | 121 | 0% | 20 | –35% | 2.8 | –63% | 0.39 | –80% |
| | DW-LKA | 13 × 13 | 1812 | +16% | 450 | +12% | 85 | –29% | 12 | –59% | 1.7 | –78% | 0.22 | –89% |
| | DW-LKA | 27 × 27 | 1812 | +16% | 490 | +22% | 56 | –54% | 7.0 | –77% | 0.87 | –89% | – | – |
| | WBMM-C | 7 × 7 | 2016 | +29% | 566 | +41% | 165 | +37% | 42 | +39% | 10.5 | +37% | 2.66 | +40% |
| | WBMM-C | 14 × 14 | – | – | **1276** | **+217%** | **242** | **+100%** | **63** | **+106%** | **15.1** | **+97%** | **3.83** | **+101%** |
| | WBMM-NC | 7 × 7 | 1582 | +1% | 525 | +31% | 161 | +33% | 42 | +38% | 10.5 | +37% | 2.67 | +40% |
| | WBMM-NC | 14 × 14 | – | – | 641 | +59% | 199 | +65% | 60 | +96% | 14.9 | +96% | 3.84 | +102% |
| B=256 | DW-Std | 3 × 3 | 4167 | +138% | 1064 | +122% | 341 | +180% | 87 | +185% | 21.8 | +185% | – | – |
| | DW-Std | 5 × 5 | 1748 | base | 480 | base | 122 | base | 31 | base | 7.6 | base | – | – |
| | DW-Std | 7 × 7 | 1073 | –39% | 286 | –40% | 73 | –40% | 18 | –40% | 4.6 | –40% | – | – |
| | DW-Std | 13 × 13 | 412 | –76% | 108 | –77% | 27 | –78% | 6.8 | –78% | 1.7 | –78% | – | – |
| | DW-Std | 27 × 27 | 120 | –93% | 29 | –94% | 7.1 | –94% | 1.7 | –94% | 0.43 | –94% | – | – |
| | DW-LKA | 7 × 7 | 2174 | +24% | 693 | +44% | 123 | +1% | 20 | –36% | 2.8 | –63% | 0.39 | – |
| | DW-LKA | 13 × 13 | 2174 | +24% | 590 | +23% | 87 | –29% | 12 | –59% | 1.7 | –78% | – | – |
| | DW-LKA | 27 × 27 | 2155 | +23% | 538 | +12% | 56 | –54% | 7.0 | –77% | 0.87 | –89% | – | – |
| | WBMM-C | 7 × 7 | 2941 | +68% | 630 | +31% | 170 | +39% | 43 | +39% | 10.5 | +38% | 2.65 | – |
| | WBMM-C | 14 × 14 | – | – | **1786** | **+272%** | **247** | **+102%** | **62** | **+102%** | **14.9** | **+96%** | **3.77** | – |
| | WBMM-NC | 7 × 7 | 2451 | +40% | 604 | +26% | 168 | +37% | 43 | +39% | 10.5 | +38% | 2.64 | – |
| | WBMM-NC | 14 × 14 | – | – | 1055 | +120% | 223 | +83% | 61 | +99% | 14.9 | +96% | 3.81 | – |

**Methods**: DW-Std = Standard depthwise convolution (implicit GEMM via PyTorch/cuDNN); DW-LKA = Large Kernel Acceleration kernels of RepLKNet/UniRepLKNet; WBMM-C = WBMM with cached weight matrix (inference mode, see Section 3.4.5); WBMM-NC = WBMM with on-the-fly weight construction (training mode). WBMM-C and WBMM-NC differ only in whether $M$ is recomputed each forward pass; they produce identical outputs. Highlighted rows show baseline ($5 \times 5$) and best WBMM configuration ($14 \times 14$ cached).

*Table 14.* Inference throughput (FPS) on ADE20K for Tiny models (NVIDIA A800 GPU, FP32). WBMM approaches Conv $3 \times 3$ speed upper bound; hierarchical variant outperforms Conv $5 \times 5$ at high resolution. Here Conv $k \times k$ denotes a depthwise convolution baseline.

| Backbone | 512×512 | | | | 512×1024 | | | | 1024×1024 | | | |
|---|---|---|---|---|---|---|---|---|---|---|---|---|
| | B2 | B4 | B8 | B16 | B2 | B4 | B8 | B16 | B2 | B4 | B8 | B16 |
| Conv $3 \times 3$ | 95.3 | 104.6 | 106.1 | 106.7 | 56.0 | 56.0 | 57.4 | 57.8 | 29.1 | 29.6 | 30.3 | 31.0 |
| Conv $5 \times 5$ | 92.4 | 101.8 | 103.3 | 104.0 | 54.4 | 54.5 | 55.7 | 56.0 | 28.1 | 28.5 | 29.2 | 29.8 |
| UniRepLKNet-T | 87.7 | 95.7 | 97.0 | 97.3 | 50.9 | 50.9 | 52.1 | 52.6 | 26.3 | 26.7 | 27.3 | 27.8 |
| UniRepLKNet-T (LKA) | 43.6 | 63.3 | 79.1 | 91.2 | 16.0 | 24.8 | 34.9 | 43.9 | 7.8 | 12.4 | 17.9 | 23.0 |
| WBMM-T ($7 \times 7$) | **90.5** | **101.6** | **103.7** | **104.6** | **54.3** | **54.9** | **56.3** | **56.5** | **28.2** | **28.8** | **29.5** | **30.1** |
| WBMM-T (Hier) | 83.7 | 96.6 | 100.3 | 102.8 | 51.8 | 53.4 | 55.7 | 56.4 | 27.7 | 28.6 | 29.6 | 30.4 |

**(3) Consistent speedup over UniRepLKNet**: WBMM delivers 1.03–1.08× speedup over UniRepLKNet (without LKA) across all tested configurations. Against LKA-accelerated UniRepLKNet, WBMM's advantage is even more pronounced at high resolution with small batch sizes where LKA's custom kernels are least efficient.

**(4) LKA severely degrades on high-resolution**: UniRepLKNet with LKA shows dramatic performance degradation on large feature maps, confirming our operator-level finding that LKA becomes counterproductive for dense prediction tasks.

### J.2. Small Model Results

Table 15 reports the throughput measurements for the Small models.

*Table 15.* Inference throughput (FPS) on ADE20K for Small models measured on a single NVIDIA A800 GPU. Conv $3 \times 3$ (a depthwise convolution baseline) represents the speed upper bound. WBMM-S+Hier uses $14 \times 14$ windows with $7 \times 7$ hierarchical parameterization.

| Backbone | 512×512 | | | | 512×1024 | | | | 1024×1024 | | | |
|---|---|---|---|---|---|---|---|---|---|---|---|---|
| | B2 | B4 | B8 | B16 | B2 | B4 | B8 | B16 | B2 | B4 | B8 | B16 |
| Conv $3 \times 3$ | 82.3 | 91.5 | 93.8 | 94.5 | 48.5 | 49.1 | 50.1 | 50.7 | 25.0 | 25.6 | 26.3 | 26.8 |
| Conv $5 \times 5$ | 78.9 | 86.9 | 89.0 | 90.2 | 46.0 | 46.6 | 47.9 | 48.6 | 24.1 | 24.4 | 25.1 | 25.6 |
| UniRepLKNet-S | 74.5 | 81.6 | 83.7 | 85.0 | 43.2 | 43.9 | 45.3 | 45.8 | 22.5 | 23.0 | 23.7 | 24.1 |
| UniRepLKNet-S (LKA) | 36.9 | 54.0 | 68.5 | 79.8 | 13.5 | 21.0 | 30.0 | 38.1 | 6.6 | 10.6 | 15.3 | 19.7 |
| WBMM-S ($7 \times 7$) | **77.7** | **88.2** | **92.2** | **93.9** | **47.4** | **48.8** | **50.3** | **50.7** | **25.0** | **25.6** | **26.3** | **26.9** |
| WBMM-S (Hier) | 72.3 | 84.5 | 89.2 | 91.5 | 44.7 | 46.8 | 48.8 | 49.5 | 23.9 | 24.8 | 25.5 | 26.3 |

**Key Observations for Small Model.** WBMM-S with $7 \times 7$ windows approaches theoretical maximum efficiency: 93.9 FPS vs 94.5 FPS for Conv $3 \times 3$ at $512 \times 512$ (only 0.6% gap), and matches Conv $3 \times 3$ exactly at $1024 \times 1024$ (26.9 vs 26.8 FPS). The hierarchical variant demonstrates strong scalability, achieving 26.3 FPS at $1024 \times 1024$ (B16) which exceeds Conv $5 \times 5$ by 2.7% despite a 7.8× larger *per-layer receptive field* (196 vs. 25 positions per layer). Compared to UniRepLKNet-S, WBMM-S achieves consistent speedup across all configurations.

## K. Hardware Generalization: CPU and Edge Device Benchmarks

To demonstrate WBMM's platform-independent efficiency, we conduct comprehensive experiments on diverse hardware configurations beyond the main GPU experiments. These results support the brief hardware generalization summary in Section 4.6.

### K.1. Experimental Setup

**Hardware Configurations:**

- **High-end GPU**: NVIDIA A800 (80GB), FP32 precision.

- **Desktop CPU**: Intel Core i7-13700K, FP32 precision.

- **Edge Device**: NVIDIA Jetson Orin Nano (8GB), GPU with FP16 precision and CPU with INT8 quantization.

## K.2. Desktop GPU and CPU Results

Table 16 reports inference speed on desktop GPU and CPU.

*Table 16.* Inference speed comparison between WBMM (W) and UniRepLKNet with/without acceleration (UA/UN) on desktop hardware. GPU: NVIDIA A800 with batch=128, FP32. CPU: Intel Core i7-13700K, FP32.

| Var. | Res. | GPU (Batch=128, FP32) | | | | | CPU (FP32) | | | | | | | | |
| | | FPS | | | Speedup | | B=1 | | | B=4 | | | B=16 | | |
| | | W | UA | UN | W/UA | W/UN | W | UN | W/UN | W | UN | W/UN | W | UN | W/UN |
|---|---|---|---|---|---|---|---|---|---|---|---|---|---|---|---|
| P | 224 | 4664 | 4484 | 3344 | 1.04× | 1.39× | 58.0 | 50.1 | 1.16× | 86.7 | 62.7 | 1.38× | 83.0 | 56.0 | 1.48× |
| | 512 | 933 | 840 | 669 | 1.11× | 1.39× | 17.0 | 11.8 | 1.44× | 12.9 | 12.5 | 1.03× | 9.7 | 7.9 | 1.23× |
| | 1024 | 230 | 180 | 166 | 1.28× | 1.39× | 2.8 | 2.3 | 1.22× | 2.2 | 1.9 | 1.16× | 1.9 | 1.7 | 1.12× |
| N | 224 | 3292 | 3174 | 2333 | 1.04× | 1.41× | 44.7 | 35.8 | 1.25× | 62.0 | 41.9 | 1.48× | 56.6 | 38.2 | 1.48× |
| | 512 | 640 | 578 | 457 | 1.11× | 1.40× | 11.9 | 8.2 | 1.45× | 9.7 | 8.3 | 1.17× | 7.9 | 6.2 | 1.27× |
| | 1024 | 159 | 124 | 113 | 1.28× | 1.41× | 2.1 | 1.7 | 1.24× | 1.8 | 1.4 | 1.29× | 1.5 | 1.3 | 1.15× |
| T | 224 | 1998 | 1963 | 1538 | 1.02× | 1.30× | 24.1 | 22.1 | 1.09× | 37.8 | 28.2 | 1.34× | 34.7 | 25.3 | 1.37× |
| | 512 | 382 | 360 | 300 | 1.06× | 1.27× | 7.2 | 5.6 | 1.29× | 6.0 | 5.6 | 1.07× | 4.8 | 4.0 | 1.20× |
| | 1024 | 96 | 80 | 75 | 1.20× | 1.28× | 1.3 | 1.1 | 1.18× | 1.1 | 0.9 | 1.22× | 0.9 | 0.8 | 1.13× |
| S | 224 | 1359 | 1341 | 1092 | 1.01× | 1.24× | 16.3 | 15.4 | 1.06× | 22.4 | 18.7 | 1.20× | 21.6 | 16.3 | 1.33× |
| | 512 | 260 | 248 | 212 | 1.05× | 1.23× | 4.5 | 3.7 | 1.22× | 3.9 | 3.7 | 1.05× | 2.9 | 2.5 | 1.16× |
| | 1024 | 65 | 56 | 53 | 1.16× | 1.23× | 0.9 | 0.7 | 1.29× | 0.7 | 0.6 | 1.17× | 0.6 | 0.5 | 1.20× |

**Key findings**: (1) On GPU, WBMM achieves 1.01–1.28× speedup over accelerated UniRepLKNet (UA) and 1.23–1.41× over non-accelerated (UN). (2) On CPU where specialized large kernel acceleration is unavailable, WBMM achieves 1.03–1.48× speedup by leveraging standard optimized matrix operations.

## K.3. Edge Device Results

Table 17 reports inference speed on the edge device.

*Table 17.* Inference speed (FPS) comparison on edge device (NVIDIA Jetson Orin Nano, 8GB). GPU uses FP16 precision; CPU uses INT8 quantization.

| Var. | Res. | GPU (FP16) | | | | | | | | | | CPU (INT8) | | | | | |
| | | Batch=4 | | | | | Batch=16 | | | | | Batch=1 | | | Batch=4 | | |
| | | W | UA | UN | W/UA | W/UN | W | UA | UN | W/UA | W/UN | W | UN | W/UN | W | UN | W/UN |
|---|---|---|---|---|---|---|---|---|---|---|---|---|---|---|---|---|---|
| P | 224 | 137 | 69.6 | 87.3 | 1.97× | 1.57× | 164 | 128 | 122 | 1.28× | 1.34× | 0.97 | 0.47 | 2.06× | 5.17 | 3.45 | 1.50× |
| | 512 | 30.7 | 9.9 | 23.1 | 3.10× | 1.33× | 32 | 21 | 23.5 | 1.52× | 1.36× | 0.66 | 0.27 | 2.44× | 1.86 | 0.92 | 2.02× |
| N | 224 | 97 | 49.1 | 77.4 | 1.98× | 1.25× | 115 | 89.6 | 84.7 | 1.28× | 1.36× | 0.73 | 0.36 | 2.03× | 4.02 | 2.71 | 1.48× |
| | 512 | 21.5 | 6.9 | 16 | 3.12× | 1.34× | 22.3 | 14.7 | 16.3 | 1.52× | 1.37× | 0.46 | 0.2 | 2.30× | 1.25 | 0.66 | 1.89× |
| T | 224 | 61.2 | 34.3 | 52.1 | 1.78× | 1.17× | 73.2 | 59.7 | 57.3 | 1.23× | 1.28× | 0.41 | 0.19 | 2.16× | 2.6 | 1.82 | 1.43× |
| | 512 | 13.6 | 5 | 10.8 | 2.72× | 1.26× | 14.2 | 10 | 11.1 | 1.42× | 1.28× | 0.25 | 0.12 | 2.08× | 0.71 | 0.31 | 2.29× |
| S | 224 | 43.9 | 26.2 | 38.1 | 1.68× | 1.15× | 51.6 | 43.3 | 41.8 | 1.19× | 1.23× | 0.22 | 0.13 | 1.69× | 1.63 | 1.23 | 1.33× |
| | 512 | 9.6 | 3.9 | 7.9 | 2.46× | 1.22× | – | – | – | – | | 0.14 | 0.08 | 1.75× | 0.41 | 0.27 | 1.52× |

**Key findings**: (1) On edge GPU with FP16, WBMM achieves 1.19–3.12× speedup over accelerated UniRepLKNet, with larger gains at higher resolutions. (2) On edge CPU with INT8 quantization, WBMM achieves 1.33–2.44× speedup, demonstrating excellent compatibility with quantization.

## K.4. Analysis and Discussion

**1. Platform-Independent Efficiency**: WBMM consistently outperforms UniRepLKNet across all tested hardware configurations because it relies on standard batched matrix multiplication operations that are universally optimized.

**2. Scaling with Input Resolution**: On GPU, WBMM's advantage increases with higher input resolution (e.g., 1.04× at 224×224 to 1.28× at 1024×1024 for WBMM-P vs UA).

**3. CPU Efficiency**: On CPU where LKA acceleration is not available, WBMM achieves substantial speedups (up to 1.48×) by leveraging highly optimized BLAS libraries.

**4. Edge Device Compatibility**: WBMM demonstrates excellent efficiency with both FP16 and INT8 precision, confirming compatibility with quantization techniques essential for edge deployment.

# L. Training Configurations

This section provides complete training configurations for all experiments reported in the main text.

## L.1. ImageNet-1K Classification

Table 18 lists the ImageNet-1K training configuration.

*Table 18.* Training configurations for ImageNet-1K experiments (FP32 precision, 8× NVIDIA A800 GPUs).

| Settings | Pico | Nano | Tiny | Small |
|---|---|---|---|---|
| Input resolution | | | $224 \times 224$ | |
| Batch size | | | 4096 | |
| Optimizer | | | AdamW | |
| Learning rate | | | 4e-3 | |
| LR schedule | | | Cosine | |
| Weight decay | | | 0.05 | |
| Warmup epochs | | | 5 | |
| Total epochs | | | 300 | |
| Mixup $\alpha$ | 0.3 | 0.5 | 0.8 | 0.8 |
| CutMix $\alpha$ | 0.3 | 0.5 | 1.0 | 1.0 |
| Random erasing | | | 0.25 | |
| Label smoothing | | | 0.1 | |
| Drop path rate | 0.1 | 0.1 | 0.2 | 0.4 |

## L.2. COCO Object Detection

Following the $3\times$ training schedule:

- Framework: MMDetection (Chen et al., 2019) with Cascade Mask R-CNN.

- Training epochs: 36.

- Batch size: 16.

- Optimizer: AdamW with learning rate 1e-4.

- Image scales: Short side 480–800, long side $\leq 1333$.

- Pre-training: ImageNet-1K weights.

## L.3. ADE20K Semantic Segmentation

- Framework: OpenMMLab semantic segmentation toolbox (MMSegmentation) with UPerNet.

- Training iterations: 160k.

- Batch size: 16.

- Optimizer: AdamW with learning rate 1e-4.

- LR schedule: Polynomial decay (power=1.0).

- Crop size: $512 \times 512$.

- Pre-training: ImageNet-1K weights.

## M. Model Architecture Specifications

Table 19 summarizes the stage depths, channel dimensions, and computational costs of all model variants.

## N. Hierarchical Reparameterization Algorithm Details

This section provides the detailed algorithms for hierarchical reparameterization, referenced from Section 3.6.

### N.1. Training Mode: Independent Multi-Scale Learning

*Table 19.* Complete model specifications showing stage depths, channel dimensions, and computational costs.

| Model | S1 Depth | S2 Depth | S3 Depth | S4 Depth | Params (M) | FLOPs (G) |
|---|---|---|---|---|---|---|
| WBMM-P | 2 | 2 | 6 | 2 | 10.6 | 1.6 |
| Channels | 64 | 128 | 256 | 512 | | |
| WBMM-N | 2 | 2 | 8 | 2 | 18.1 | 2.7 |
| Channels | 80 | 160 | 320 | 640 | | |
| WBMM-T | 3 | 3 | 18 | 3 | 31.0 | 4.8 |
| Channels | 80 | 160 | 320 | 640 | | |
| WBMM-S | 3 | 3 | 27 | 3 | 55.6 | 9.0 |
| Channels | 96 | 192 | 384 | 768 | | |

*Table 20.* Stage-specific block patterns. "W" denotes WBMM $7 \times 7$ block; "D" denotes $3 \times 3$ depthwise convolution; "↔" indicates alternating pattern.

| Model | S1 | S2 | S3 | S4 | Task |
|---|---|---|---|---|---|
| Pico | [D,D] | [W,D] | [W↔D]×3 | [W,W] | Classification |
| Nano | [D,D] | [W,D] | [W↔D]×4 | [W,W] | Classification |
| Tiny | [D,D,D] | [W,D,W] | [W↔D]×9 | [W,W,W] | Classification |
| Small | [D,D,D] | [W,D,W] | [W,D,D]×9 | [W,W,W] | Classification |
| Tiny* | [W,D,W] | [W,D,W] | [W↔D]×9 | [W,W,W] | Seg./Det. |
| Small* | [W,D,W] | [W,D,W] | [W,D,D]×9 | [W,W,W] | Seg./Det. |

*For dense prediction variants, S1 and S2 use hierarchical $14 \times 14 + 7 \times 7$ windows, S3 uses a single $14 \times 14$ window, and S4 uses a single $7 \times 7$ window (its feature map being the smallest in the network).

---

**Algorithm 2** WBMM Training Mode with Hierarchical Parameterization

---

**Require:** Input $X \in \mathbb{R}^{B \times C \times H \times W}$
**Require:** Global bias $R_g \in \mathbb{R}^{C \times 27^2}$ for $14 \times 14$ windows
**Require:** Local bias $R_l \in \mathbb{R}^{C \times 13^2}$ for $7 \times 7$ sub-windows
 1:
 2: **// Global scale:** $14 \times 14$ **windows (with identity shortcut)**
 3: $X_w \leftarrow$ WindowPartition$(X, 14, 14)$
 4: $M_g \leftarrow$ ConstructMatrix$(R_g)$
 5: $Y_g \leftarrow X_w \cdot M_g + X_w$     *// residual / identity shortcut*
 6:
 7: **// Local scale:** $7 \times 7$ **sub-windows (with identity shortcut)**
 8: $X_{sub} \leftarrow$ Repartition$(X_w, [C, N, 2, 7, 2, 7])$
 9: $M_l \leftarrow$ ConstructMatrix$(R_l)$
10: $Y_l \leftarrow X_{sub} \cdot M_l + X_{sub}$     *// residual / identity shortcut*
11: $Y_l \leftarrow$ ReshapeToGlobalWindows$(Y_l)$
12:
13: **// Additive fusion**
14: $Y_w \leftarrow Y_g + Y_l$
15: $Y \leftarrow$ InversePartition$(Y_w, H, W)$
16: **return** $Y$

---

---

**Algorithm 3** WBMM Inference Mode with Fused Hierarchical Matrix

---

**Require:** Trained $R_g \in \mathbb{R}^{C \times 27^2}$, $R_l \in \mathbb{R}^{C \times 13^2}$
1:
2: **// One-time fusion at model load**
3: $M_g \leftarrow$ ConstructMatrix($R_g$)  // shape $(C, 196, 196)$ *for* $14 \times 14$ *windows*
4: $M_l \leftarrow$ ConstructMatrix($R_l$)  // shape $(C, 49, 49)$ *for* $7 \times 7$ *sub-windows*
5: *// view* $M_g$ *with 9 axes:* $(C, s_{out}^{row}, p_{out}^{row}, s_{out}^{col}, p_{out}^{col}, s_{in}^{row}, p_{in}^{row}, s_{in}^{col}, p_{in}^{col})$,
6: *// where* $s^* \in \{0, 1\}$ *selects one of* $2 \times 2$ *sub-windows;*
7: *//* $p^* \in \{0, \ldots, 6\}$ *is the position inside that* $7 \times 7$ *sub-window*
8: $M_g \leftarrow M_g$.view$(C, 2, 7, 2, 7, 2, 7, 2, 7)$
9: $\tilde{M_l} \leftarrow M_l + I_{49}$  *// absorb local identity shortcut into* $M_l$
10: $\tilde{M_l} \leftarrow \tilde{M_l}$.view$(C, 7, 7, 7, 7)$  *// match a single diagonal block of* $M_g$
11: **for** $i = 0$ to $1$ **do**
12:  **for** $j = 0$ to $1$ **do**
13:    $M_g[:, i, :, j, :, i, :, j, :] \leftarrow M_g[:, i, :, j, :, i, :, j, :] + \tilde{M_l}$  *//* $s_{out}^{row} = s_{in}^{row} = i$, $s_{out}^{col} = s_{in}^{col} = j$: *only diagonal blocks updated*
14:  **end for**
15: **end for**
16: $M_{\text{fused}} \leftarrow M_g$.view$(C, 196, 196) + I_{196}$  *// absorb global identity shortcut*
17:
18: **// Forward pass (zero overhead)**
19: $X_w \leftarrow$ WindowPartition$(X, 14, 14)$
20: $Y_w \leftarrow X_w \cdot M_{\text{fused}}$
21: $Y \leftarrow$ InversePartition$(Y_w, H, W)$
22: **return** $Y$

---

### N.2. Inference Mode: Zero-Cost Matrix Fusion

**Zero overhead property.** At deployment, the merged $M_{\text{fused}}$ is a single $C \times 196 \times 196$ tensor, so spatial mixing is carried out by one batched matrix multiplication over $14 \times 14$ windows—no more arithmetic per block than single-scale WBMM at $w = 14$, while expressing both global and local patterns. The two tables $R_g$ and $R_l$ are learned independently and both remain as parameters of the deployed model; their fusion is a one-time tensor addition at load and is invisible to the forward pass.

**Training–inference equivalence.** Algorithms 2 and 3 are functionally identical: the training mode computes $Y_w = X_w M_g + X_w + X_{sub} M_l + X_{sub}$ (with $X_{sub} M_l + X_{sub}$ reshaped back to global windows), while the inference mode computes $Y_w = X_w M_{\text{fused}}$. Because $X_{sub}$ is just $X_w$ repartitioned into $2 \times 2$ sub-windows, the contribution $X_{sub} M_l$ is structurally equivalent to multiplying $X_w$ by a block matrix that contains $M_l$ on its four diagonal blocks (and zeros off-diagonal), and the two identity shortcuts collapse into the $196 \times 196$ identity. Therefore $M_{\text{fused}} = M_g + \text{diag\_blocks}(M_l, M_l, M_l, M_l) + 2 I_{196}$ reproduces the training-mode output exactly.

Per the segmentation/detection architectural specification (Table 20), this hierarchical fusion is used at the high-resolution stages S1 and S2 where multi-scale context contributes most; deeper stages run a single window per block, with S3 operating at $14 \times 14$ for spatial coverage and S4 using a single $7 \times 7$ window at the network's smallest feature map.

## O. Limitations and Future Work

We note WBMM's current limitations along several dimensions.

**Memory footprint at very large windows.** The constructed weight matrix $M \in \mathbb{R}^{C \times d \times d}$ grows as $O(C \cdot w^4)$ with window size $w$. At $w = 14$ ($d = 196$), $M$ takes on the order of tens of MB across the network's stages (specifically, $\sim$37.5 MB per 256-channel layer at FP32, lower for earlier stages with fewer channels)—negligible compared to activations. For windows beyond $28 \times 28$, however, materializing the full $M$ becomes memory-prohibitive. Crucially, the bias table $R$ remains compact ($O(C \cdot (2w-1)^2)$), so this is an engineering rather than a representational limitation: a Flash-Attention-style tiling kernel that stores $R$ in SRAM and computes $M$ blocks on-the-fly via the index map (Equations (7) and (8)) would avoid full materialization entirely. Because WBMM is purely linear (no softmax), such tiling is strictly simpler than Flash Attention's online softmax. Triton-based tiled kernels are a planned engineering effort.

**Window size vs. task.** Pure large windows are not uniformly optimal: Table 4 shows that pure $14 \times 14$ underperforms pure

$7 \times 7$ on ADE20K. The hierarchical reparameterization of Section 3.6 resolves this by combining global and local patterns without slowing inference, but task-specific window selection still requires design effort. Automated window-size search is a natural next step.

**Input independence.** WBMM constructs a single weight matrix shared across all samples and windows. This is a feature for efficiency on dense prediction (Section D: WBMM matches or beats parameter-matched window attention on segmentation while being noticeably faster), but content-adaptive routing tasks may benefit from input-dependent weights. A lightweight input-dependent gate $\sigma(\cdot)$ over the cached $M$ is a promising hybrid we leave to future work.

**Generalization to 1D and 3D data.** This paper focuses on 2D vision. WBMM's core abstraction—a compact relative-position bias table indexed into a per-channel weight matrix, applied via batched MatMul to contiguous windows—naturally extends to 1D temporal sequences and 3D spatio-temporal video. The optimal window shape (e.g., $T \times H \times W$ or factorized variants) and cross-window strategies in these domains require dedicated study and constitute a promising future direction.

