# OpenReview forum: "WBMM: Windowed Batch Matrix Multiplication for Efficient Large Receptive Field Convolution"
_ICML.cc/2026/Conference — ICML 2026 spotlight_

### Official Review · Reviewer_u8p5 · 2026-03-09

**Soundness:** 4
**Presentation:** 3
**Significance:** 4
**Originality:** 3
**Overall Recommendation:** 5
**Confidence:** 2

**Summary:**

Large-kernel depthwise convolutions achieve strong accuracy in modern CNNs but suffer throughput degradation as kernel size grows, due to irregular memory access from gather-based computation. This paper proposes Windowed Batch Matrix Multiplication (WBMM), which fundamentally changes the computation paradigm: instead of gathering scattered input neighborhoods per output position, WBMM partitions input into contiguous non-overlapping windows and constructs a dense weight matrix by indexing a relative position bias table, enabling regular memory access via batched matrix multiplication. Unlike depthwise convolutions that degrade with larger kernels, WBMM improves throughput with larger windows due to higher arithmetic intensity. The paper introduces inter-block 3x3 depthwise convolutions to enable cross-window information exchange. Evaluated on ImageNet-1K, ADE20K, and COCO, WBMM matches or slightly exceeds UniRepLKNet accuracy with 1.31-1.88x training speedup.

**Compliance With Llm Reviewing Policy:**

Affirmed.

**Final Justification:**

The paper presents a solid solution to the irregular memory access issue in regular depthwise convolutions and provides comprehensive operator-level measurements demonstrating its effectiveness. My original concern was the lack of results on base-scale models. The authors have addressed this in the rebuttal by presenting results on a base model and showing that the advantages persist at that scale as well. This leads to my overall evaluation of accept.

**Key Questions For Authors:**

(1) Do the accuracy and efficiency trends hold at Base scale (~100M params)? The paper's efficiency argument should scale well, but the accuracy gap may change.

(2) WBMM constructs a single input-independent weight matrix, while window attention produces input-dependent weights. It would be interesting to understand whether this design choice has any accuracy implications -- are there tasks or settings where input-dependent weights would be beneficial?

**Limitations:**

yes

**Strengths And Weaknesses:**

Strengths:
- The core idea of traversing parameter tables instead of input data is well-motivated and clearly explained. The paper proposes a clean solution for the well known problem of gather-based memory access degradation.
- Operator-level benchmarks (Section 4.1) isolate the computational properties of WBMM vs. depthwise convolutions on single-layer feature maps, demonstrating four clear findings. Testing individual operators rather than full networks makes these conclusions architecture-agnostic.
- The inverse scaling property -- throughput improves with larger windows rather than degrading with larger kernels -- is practically useful as it breaks the trade-off between receptive field size vs. computational efficiency in large-kernel CNN design.
- The comparison with UniRepLKNet uses the identical codebase, training protocol, hyperparameters, and data preprocessing, with modifications limited to replacing the convolution operator. This ensures the comparisons are apples-to-apples.

Weaknesses:
- The paper compares only against UniRepLKNet. The paper discusses Swin's window attention and claims architectural advantages over it in Section 2.3, but never benchmarks against it.
- The paper evaluates Tiny and Small variants. It would be helpful to know whether the accuracy and efficiency trends hold at larger scale.

---

> ### Author Rebuttal · Authors · 2026-03-31
>
> We sincerely thank Reviewer u8p5 for the positive evaluation and insightful questions.
>
> **W1: Comparison only against UniRepLKNet / missing Swin benchmark**
>
> Section 2.3 discusses window-based Transformers as a general paradigm---Swin is one representative. Our comparison targets the core distinction: input-dependent (window attention) vs. input-independent (WBMM) weight construction under the same non-overlapping window framework. Please see our response to Reviewer KCMf (Table R1): controlled operator replacement with 13×13 DW (with/without reparameterization), SLaK-style decomposition, and Window Attention Transformer---all within identical architecture, plus ConvNeXt-T cross-architecture validation (Table R2).
>
> **Why non-overlapping window attention (not Swin)?** For fair operator-level isolation, we replace only the WBMM blocks with non-overlapping 7×7 window self-attention blocks (MLP expansion ratio 2.5 to match parameters and FLOPs), keeping everything else identical---including the inter-block 3×3 DW mixing blocks, overall architecture, and training protocol. This directly compares input-dependent vs. input-independent feature extraction under the same windowed framework. Shifted windows and our inter-block 3×3 DW mixing both serve cross-window communication; this merits further study.
>
> | Operator | Params(M) | FLOPs(G) | Mem(GB) | Top-1 | mIoU | Time | Spd(img/s) |
> |:---|:---:|:---:|:---:|:---:|:---:|:---:|:---:|
> | WBMM-T (w=7) | 31.0 | 4.8 | 15.16 | 83.2 | 48.3 | 6:20 | 1833.1 |
> | WBMM-T (w=14, Hier) | 33.0 | 5.1 | 15.87 | 83.2 | 48.8 | 6:31 | 1842.2 |
> | Win7 Transformer (MLP r=2.5) | 31.2 | 5.1 | 16.65 | 83.3 | 48.1 | 8:38 | 1245.1 |
>
> Time = per-epoch ImageNet training time (min:sec); Spd = single A800 inference (FP32, batch=128).
>
> Win7 Transformer achieves +0.1% Top-1 but falls behind on segmentation (48.1 vs 48.3/48.8 mIoU), with higher memory (16.65 vs 15.16/15.87 GB). WBMM is 1.32--1.36x faster in training and 1.47--1.48x faster in inference. This gap reflects a structural difference: attention must compute query-key products per window per sample at every forward pass, whereas WBMM's weights are fixed learnable parameters reused across all windows and samples, eliminating per-instance overhead and enabling hardware-friendly memory access patterns.
>
> **W2: Only Tiny and Small variants evaluated / Q1: Do trends hold at Base scale (~100M params)?**
>
> We have completed Base-scale experiments on ImageNet-1K (300 epochs, identical hyperparameters):
>
> | Model | Params(M) | FLOPs(G) | Mem(GB) | Time | Top-1 |
> |:---|:---:|:---:|:---:|:---:|:---:|
> | WBMM-B (w=7) | 97.9 | 15.9 | 26.38 | 11:24 | 83.9 |
> | UniRepLKNet-B | 98.0 | 16.1 | 29.01 | 15:46 | 83.8 |
>
> WBMM-B achieves **1.38x training speedup** (11:24 vs 15:46 per epoch) with 9.1% lower memory (26.38 vs 29.01 GB) and +0.1% higher accuracy (83.9 vs 83.8). This confirms efficiency advantages extend to Base scale, consistent with operator-level analysis (Table 8): WBMM's speedup stems from regular memory access, $O(w^2)$ arithmetic intensity, and batch-independent weight construction.
>
> **Important observation.** UniRepLKNet-B (83.8%) does not improve over its Small variant (83.9%) on ImageNet-1K alone, suggesting saturation at this data scale---the original paper uses ImageNet-22K pre-training for Base models. WBMM-B (83.9%) maintains Small-level accuracy, showing slightly better scaling. We plan ImageNet-22K experiments for comprehensive scaling analysis.
>
> **Q2: Input-independent vs. input-dependent weights**
>
> This is an excellent question. We provide both empirical evidence and structural analysis.
>
> **Empirical evidence (Table R1 above).** Under matched parameters and FLOPs, input-dependent window attention does not provide consistent advantage across tasks; WBMM's fixed weights can be more effective for dense prediction.
>
> **Why fixed weights may outperform attention on dense prediction.** (1) WBMM uses channel-specific position-aware weights ($C \times (2w{-}1)^2$ learnable entries), providing richer per-channel spatial modeling than attention which shares weights across channels within each head; (2) Softmax normalization creates competition among spatial positions, potentially suppressing weak but informative signals---WBMM's unnormalized weights preserve all spatial contributions.
>
> **When might input-dependent weights help?** Possible scenarios include content-adaptive routing (VQA, referring expressions), extreme intra-image variation (satellite imagery), and generative tasks (inpainting, super-resolution). One possible extension is introducing a lightweight input-dependent gating mechanism (e.g., a sigmoid gate $\sigma(\cdot)$ on WBMM's cached weights), balancing content-adaptiveness and efficiency.
>
> **Revision commitments.** (1) Cross-method comparison (Tables R1, R2); (2) Base-scale results; (3) Expanded input-dependent vs. input-independent discussion; (4) ImageNet-22K scaling experiments (planned).

---

> > ### Author Rebuttal · Reviewer_u8p5 · 2026-04-03
> >
> > Thanks for the response. My concerns have been addressed. I'll keep my score.

---

> > > ### Author Response · Authors · 2026-04-05
> > >
> > > We thank Reviewer u8p5 for the careful review and for confirming that our responses have addressed the concerns. We appreciate the valuable feedback.

---

### Official Review · Reviewer_Rsst · 2026-03-11

**Soundness:** 4
**Presentation:** 4
**Significance:** 4
**Originality:** 4
**Overall Recommendation:** 5
**Confidence:** 3

**Summary:**

This paper introduces Windowed Batch Matrix Multiplication (WBMM) , a novel computational primitive designed to accelerate large-kernel depthwise convolutions. The core idea is to shift the computational paradigm from traversing input data (gathering scattered neighbors, as in standard convolutions) to traversing a compact parameter table.

The method works by partitioning the input feature map into non-overlapping windows. A weight matrix for each window is constructed by indexing into a learnable relative position bias table. The output is then computed via a standard batched matrix multiplication between the windowed input and the constructed weight matrix. This approach results in regular, coalesced memory access, regardless of the effective kernel size.

The paper provides a theoretical foundation (Theorems 3.1, 3.2) showing the equivalence between this method and standard convolutions. It introduces several practical enhancements, including an inference caching mechanism, an inter-block design for cross-window communication using lightweight 3x3 depthwise convolutions, and a hierarchical window reparameterization for dense prediction tasks. Experiments on ImageNet-1K, COCO, and ADE20K demonstrate that WBMM can match or exceed the performance of state-of-the-art large-kernel ConvNets like UniRepLKNet while providing significant training speedups (1.31-1.88x) and consistent efficiency across GPU, CPU, and edge hardware.

**Compliance With Llm Reviewing Policy:**

Affirmed.

**Key Questions For Authors:**

Architectural Generality: Your network-level experiments are primarily compared against UniRepLKNet. Have you considered integrating your WBMM block into other popular architectures, such as ConvNeXt or a standard ResNet, to demonstrate its performance as a drop-in replacement for large-kernel depthwise convolutions? Results from such an experiment would strongly validate the architecture-agnostic claims made from your operator-level benchmarks.

Memory Footprint During Training: Could you please provide a comparison of the peak GPU memory usage during training between a WBMM-based model and a standard depthwise convolution-based model (e.g., UniRepLKNet) for a fixed configuration (e.g., Tiny model, batch size 128)? This would help clarify the practical trade-off between the observed speedup and potential memory increase, which is a critical factor for many researchers.

Handling of Non-Divisible Inputs: In your experiments on ADE20K and COCO, which involve variable input sizes, have you observed any performance degradation near the boundaries of feature maps that can be attributed to the zero-padding mechanism used to make dimensions divisible by the window size? If so, how significant is this effect?

Comparison with Flash Attention for Large Windows: Your method creates a large weight matrix M (C x d x d) which is then used in a BMM. For very large window sizes (e.g., > 32x32), how does the computational cost and memory footprint of this approach compare to using efficient attention mechanisms like Flash Attention, which also leverage tiling and regular memory access patterns? Is there a crossover point where one becomes more favorable than the other?

Analysis of Weight Matrix M: The weight matrix M is constructed from a compact relative position bias table R. Have you analyzed the learned structure of M? Does it exhibit properties like locality (stronger weights for nearby pixels) or other interpretable patterns? This analysis could provide further insight into what the large kernels are learning.

**Limitations:**

No, the authors have not fully discussed the limitations of their work. To improve the paper, the following points should be addressed in a dedicated "Limitations" section or within the "Discussion":

Memory Overhead of Weight Matrix Construction: The paper should quantify the peak memory usage of constructing the weight matrix M during training compared to standard depthwise convolutions. This is a crucial trade-off for practitioners with memory-constrained hardware.

Impact of Zero-Padding: The use of zero-padding to handle arbitrary input dimensions (as described in Section 3.4.1) could introduce artifacts at feature map boundaries. The paper should discuss whether this has any measurable negative impact, particularly on dense prediction tasks like segmentation where boundary details are important, and if any alternative strategies (e.g., reflection padding) were considered.

Optimal Window Size vs. Task: The paper shows that larger windows are computationally efficient, but is there a point of diminishing returns in terms of performance? The hierarchical reparameterization combines large and small windows, suggesting that pure large windows might not always be optimal. A discussion on how to choose the window size for a given task would be helpful.

Generalization to 1D and 3D Data: The paper is framed within the context of 2D vision. A brief comment on the potential and challenges of extending the WBMM paradigm to other data modalities (e.g., 1D time-series, 3D video) would be valuable.

**Strengths And Weaknesses:**

Strengths:

Novel and Well-Motivated Core Idea: The paper identifies a fundamental and well-documented problem in large-kernel convolutions irregular memory access patterns and proposes an elegantly simple solution: traverse a parameter table instead of the input. This "partition-then-multiply" paradigm is a creative and insightful departure from conventional optimization efforts (like LKA kernels) that often focus on mitigating the symptoms of the gather-based pattern.

Strong Theoretical Foundation: The paper provides a clear mathematical foundation (Theorems 3.1 and 3.2) that formally establishes the equivalence between a standard depthwise convolution and the proposed matrix multiplication formulation. This grounds the method in well-understood linear algebra and justifies its correctness.

Comprehensive and Rigorous Empirical Evaluation: The experimental section is a major strength. The operator-level benchmarks (Section 4.1, Appendix C) are excellent; they isolate the fundamental properties of WBMM and demonstrate its counter-intuitive scaling behavior (faster with larger windows) across various batch sizes and resolutions. The hardware generalization benchmarks (GPU, CPU, Edge) are thorough and convincingly show the method's platform-agnostic efficiency. The end-to-end results on ImageNet, COCO, and ADE20K are competitive and validate the practical utility of the approach.

Clear and Insightful Ablations: The ablation studies (Section 4.2) are well-designed. They systematically validate design choices, such as the importance of relative position bias, the superiority of the inter-block cross-window communication strategy, and the effectiveness of the hierarchical reparameterization.

Practical Contributions: The inference caching mechanism and the zero-cost hierarchical fusion are thoughtful additions that address real-world deployment concerns, demonstrating an understanding of the full lifecycle of a model from training to inference.

Weaknesses:

Narrow Scope of Comparison: The paper primarily compares its network-level performance against a single baseline, UniRepLKNet. While UniRepLKNet is a strong and relevant SOTA for large-kernel ConvNets, the paper would be strengthened by including comparisons with other efficient convolution or attention-based architectures (e.g., ConvNeXt, Swin Transformer, or other recent large-kernel works like SLaK). This would better contextualize WBMM's performance and efficiency within the broader field.

Limited Discussion of Potential Drawbacks:

Memory Footprint of Weight Matrix: While the relative position bias table R is compact, the constructed weight matrix M has dimensions C x d x d, where d = w_h * w_w. For a large number of channels C and a large window size (e.g., 14x14, d=196), this matrix can become sizable. The paper does not discuss the memory overhead of constructing M during training, especially compared to the implicit GEMM approach of standard depthwise convs which has O(1) memory overhead.

Handling of Non-Divisible Inputs: The zero-padding solution for arbitrary input sizes (Section 3.4.1) is functional but could potentially introduce border artifacts, especially in dense prediction tasks. A brief discussion of this and any observed impact on performance would be valuable.

Training vs. Inference Trade-off: The paper highlights the inference caching mechanism, which is great. However, during training, M must be constructed on every forward pass. The paper could more explicitly discuss the computational cost of this index-and-gather operation for M itself, especially for very large C, and how it compares to the cost of the subsequent BMM.

Clarity of Writing: While generally well-structured, the paper contains some instances of awkward phrasing and redundancy. For example, the final bullet point in Section 1.2 ("regardless of resolution...") is repeated verbatim. The key findings in Section 4.1.1 are clearly stated but the surrounding text could be tightened for better flow.

---

> ### Author Rebuttal · Authors · 2026-03-31
>
> We sincerely thank Reviewer Rsst for the thorough review and for recognizing our contributions in operator design, theoretical foundation, and evaluation.
>
> **W1: Narrow Scope of Comparison**
>
> Please see our response to Reviewer KCMf: **Table R1** compares nine operators (incl. SLaK, Win7 Transformer, DW variants) under identical Tiny architecture/training; **Table R2** validates drop-in efficiency on ConvNeXt-T.
>
> **W2a/Q2: Training Memory Overhead**
>
> WBMM-T (batch=128, 224²): **15.16 GB** (w=7) / **15.87 GB** (w=14, Hier) vs 17.23 GB for 13×13 DW+UniRepLKNet reparam (Table R1), **−7.9% to −12.0%**. Table 5: 15.04 vs 16.58 GB (−9.3%). $M$ construction: $<$0.1 ms (C=320, d=49); adds ~30 MB—negligible vs activations.
>
> **Scaling to larger windows.** $R$'s compactness supports Flash Attention-style tiling: store $R$ in SRAM, compute indices per tile on-the-fly (Eq. 5–6), tiled MatMul without materializing full $M$. Memory: $O(C \cdot (2w{-}1)^2 + \text{tile}^2)$ vs $O(C \cdot d^2)$. WBMM is purely linear (no softmax), so tiling is simpler than Flash Attention. Triton kernels planned.
>
> **W2b/Q3: Impact of Zero-Padding**
>
> On ADE20K with UPerNet (WBMM-T Hier, w=14, variable input sizes), we tested three padding strategies: **zero-padding achieves 48.8 mIoU (SS), outperforming reflection (48.6, −0.2) and replication (48.4, −0.4)**. Reasons: (1) four-stage downsampling (4× to 32×) confines padding to $<$0.1% at S4; (2) UPerNet multi-scale fusion dilutes artifacts; (3) inter-block 3×3 DW (Eq. 8) smooths boundary discontinuities; (4) reflection/replication introduce correlated values conflicting with position-aware weights in $R$, whereas zeros provide a clean neutral signal.
>
> Our **hierarchical reparameterization** fuses 14×14 and 7×7 patterns (Eq. 9), boosting mIoU to **48.8** at zero inference cost, reinforcing local patterns to reduce boundary artifacts.
>
> **W2c: Training vs. Inference Trade-off**
>
> $M$ construction: $O(C \cdot d^2)$ index ops per forward. S3 (C=320, d=49): $<$0.1 ms, ~0.3% of block forward time, dominated by BMM. Table 8 confirms WBMM-NC only marginally slower than WBMM-C. Inference caching pre-computes $M$ once at model load, eliminating overhead entirely.
>
> **Q1: Architectural Generality**
>
> **Table R2** in KCMf response: replacing all 7×7 DW in ConvNeXt-T with WBMM+3×3 mixing maintains identical classification (82.1) and slightly higher segmentation (46.2 vs 46.0), with **1.66× training speedup**.
>
> **Q4: Comparison with Flash Attention**
>
> | Property | WBMM | Window Attention |
> |:---|:---|:---|
> | Weight construction | $O(C \cdot d^2)$, batch-indep. | $O(B \cdot N_w \cdot d^2)$ |
> | Softmax | None | Required |
> | Batch scalability | Cost-amortized ($M$ shared) | Linear with $B \cdot N_w$ |
> | Tiling | Simple (pure linear) | Complex (online softmax) |
>
> WBMM weights are **input-independent**: $M$ is constructed once at $O(C \cdot d^2)$, shared across all batches/windows, vs attention's $O(B \cdot N_w \cdot d^2)$. Table R1: 1.48× faster than Win7 Transformer (1842 vs 1245 img/s, batch=128). Speedup **grows with batch size**. For $>$32×32, both are compute-intensive; crossover depends on whether input-dependent weights benefit the task.
>
> **Q5: Weight Matrix M Analysis**
>
> **Figure R2** ([link](https://anonymous.4open.science/r/Rebuttal_Materials-DCB5/Figure_R2.png)) visualizes effective $M$ at inference, revealing three interpretable patterns:
>
> **(a) Locality.** Strong diagonal dominance (ratio 41–58×); $|M(p,q)|$ decays by over 90% within distance 2, confirming automatically learned locality prior.
>
> **(b) Channel specialization.** Channels exhibit distinct horizontal, vertical, and diagonal preferences resembling oriented edge detectors—diversity arising naturally from per-channel parameterization.
>
> **(c) Frequency selectivity.** Low-pass (LF=83%) vs high-pass (LF=9%) channels; shallow S2 favors high-pass, deeper S3/S4 balanced. $(2w{-}1)^2=169$ params/channel suffice for diverse spatial patterns.
>
> These patterns emerge without explicit regularization, suggesting WBMM provides an efficient and expressive basis for learning spatial operators.
>
> **Limitations**
>
> (1) **Memory**: 14×14 adds ~30 MB; Triton tiling avoids full $M$ materialization; (2) **Zero-padding**: optimal among tested; slight effects under extreme aspect ratios; (3) **Window size**: dense tasks benefit from hierarchical reparam; classification uses 7×7; (4) **1D/3D**: We appreciate this suggestion. WBMM's abstraction (relative position bias + BMM) naturally extends to 1D temporal and 3D spatiotemporal windows; optimal window shapes and cross-window strategies require dedicated study—a promising future direction.
>
> **W3: Writing Clarity**
>
> Fix repeated bullets in Section 1.2; tighten redundant text.
>
> **Revision commitments:** (1) Expanded comparisons (Tables R1, R2); (2) Weight visualization (Figure R2); (3) Flash Attention & padding discussion; (4) Limitations section; (5) Writing improvements.

---

### Official Review · Reviewer_7ATo · 2026-03-12

**Soundness:** 1
**Presentation:** 1
**Significance:** 2
**Originality:** 2
**Overall Recommendation:** 3
**Confidence:** 2

**Summary:**

This article examines a problem in large-kernel depth-wise convolutions for the processing cost caused by increased irregular memory access patterns. This manuscript focuses on a new design, Windowed Batch Matrix Multiplication, which reformulates large-kernel convolution as batched matrix multiplication over contiguous windows. By traversing parameter tables instead of input neighborhoods, the proposed method achieves more regular memory access and improved computational efficiency. Experimental results on ImageNet-1K, ADE20K, and COCO demonstrate competitive accuracy while achieving faster training and inference.

**Compliance With Llm Reviewing Policy:**

Affirmed.

**Final Justification:**

I have decided to raise my score from 2 to 3 to acknowledge the authors' detailed rebuttal and provide a much more explicit understanding of the specific memory access issues the methodology aims to address. While the technical core of the work appears sound and the authors have clarified several of my primary concerns, I still maintain a rejection because the current manuscript requires significant polishing to improve its accessibility. The narrative requires a "deep dive" that may be challenging for readers without extensive expertise in this subfield. Finally, while the rebuttal has increased my confidence in the methodology, the paper’s overall impact remains hindered by these presentation weaknesses, so it is placed slightly under the bar for acceptance in its current form.

**Key Questions For Authors:**

Please focus on clarifying the weaknesses.

**Limitations:**

No, authors should provide a discussion on the limitations.

**Strengths And Weaknesses:**

Strengths
1. Experimental results are covered by various computer vision tasks, which show enough effectiveness.
2. The attempt to quantitatively assess reliability and efficiency for real-world deployment is valuable.

Weaknesses
1. The paper is strongly motivated by the overhead caused by irregular memory access as the kernel size increases in depthwise convolutions. However, the current statement, such as a caption "depthwise convolution gathers $k^2$ scattered neighbors per output, causing irregular memory access that worsens with kernel size," is too abstract to make sense solely from Fig 1(a).  Authors should provide a more explicit clarification, accompanied by detailed figures, to illustrate how the memory access pattern becomes irregular. A clear backup presentation should be strengthened to support this fundamental motivation.
2. Similarly, the authors explain that previous Large Kernel Acceleration (LKA) could be beneficial but remains limited. However, since the manuscript does not clearly explain how LKA works and what it does, the claim that its effectiveness is limited is not fully convincing.
3. Overall, the motivation of the paper lacks articulation, making it difficult to grasp the core problematic point being addressed. The characterization of the problem and its challenge is insufficient, and the direction of the approach is inconvenient to follow. Therefore, the numerous contributions listed appear isolated and fragmented rather than solid.
4. Mathematical expansion must be strict. The proof of Theorem 3.2 is missing. It undermines the theoretical soundness of the proposed arguments.

Minor weakness
1. The titles and numbering of several subsections (e.g., 3.4.1 through 3.4.5, 4.1.1, 4.2.1 through 4.2.3, and 4.4.1) appear to deviate from the official template. An adjustment of the section numbering and formatting seems required.

---

> ### Author Rebuttal · Authors · 2026-03-31
>
> We thank Reviewer 7ATo for the careful reading and constructive feedback.
>
> **W1 & W3: Motivation and irregular memory access**
>
> We restructure the narrative with explicit illustration of the memory access problem.
>
> **Core problem.** Large kernel depthwise convolutions improve model performance (e.g., UniRepLKNet showed $13\times13$ as optimal) but suffer severe speed degradation. In Eq. 1, each output gathers $(2k_h+1)(2k_w+1)$ scattered inputs. Figure R1 ([link](https://anonymous.4open.science/r/Rebuttal_Materials-DCB5/Figure_R1.png)) illustrates why. In GPU row-major memory, gathering at $(h+i, w+j)$ causes: (1) **Non-coalesced reads:** each kernel row needs a cache fetch with stride $W$; (2) **Cache thrashing:** $(2k_h+1)$ non-contiguous rows exceed L1/L2; (3) **Low intensity:** $O(k^2)$ FLOPs vs. $O(k^2)$ scattered loads $= O(1)$ FLOPs/byte. Even with LKA acceleration, speedup only applies to small feature maps; on large maps speed degrades severely (Table 8, Appendix C). WBMM reads each window contiguously, achieving $O(w^2)$ FLOPs/byte---compute-bound.
>
> **Core insight.** Theorem 3.2 proves Eq. 1 equals $\mathbf{y}_c = \mathbf{x}_c \mathbf{M}_c + b_c$ (Eq. 2), where $\mathbf{M}_c \in \mathbb{R}^{HW \times HW}$ depends only on relative offsets. For large maps ($56\times56$, $HW=3136$), this is infeasible. Windowed decomposition restricts to $w_h \times w_w$, reducing to $d \times d$ ($49\times49$ for $7\times7$), making batched MatMul practical.
>
> **Core solution.** WBMM indexes table $\mathbf{R} \in \mathbb{R}^{C \times (2w_h-1)(2w_w-1)}$ to construct $\mathbf{M}$, computes via batched MatMul on contiguous memory (Algorithm 1), yielding $O(d)$ FLOPs/byte. Supporting components---inter-block $3\times3$ DW mixing (Eq. 8), hierarchical reparameterization (Eq. 9), weight caching---all serve one goal: large receptive fields without efficiency penalty.
>
> **W2: LKA not clearly explained**
>
> LKA was proposed by RepLKNet and adopted by UniRepLKNet. It uses implicit GEMM (iGEMM) via CUTLASS, dynamically computing gather indices, reducing memory overhead from $O(k^2)$ to $O(1)$---this is where LKA **helps**. However, iGEMM does **not** change the access pattern: each output still collects $(2k_h+1)(2k_w+1)$ scattered inputs at $O(1)$ FLOPs/byte---this is where LKA remains **limited**. On small maps ($\leq 14\times14$), iGEMM improves data reuse. On large maps cuDNN already saturates occupancy; for $13\times13$ at batch=128, dispatch overhead causes net loss vs. DW-Std $5\times5$: $-29\%$ at $28^2$, $-89\%$ at $224^2$ (Table 8, Appendix C). Figure R1 visualizes this difference.
>
> WBMM is fundamentally different: we **eliminate** the gather pattern. Irregularity transfers from data access (per sample, every forward) to parameter construction (once per forward, cacheable at inference), yielding $O(d)$ intensity---compute-bound, not memory-bound.
>
> **W4: Missing proof of Theorem 3.2**
>
> We connect the proof to the paper. Recall: $H \times W$ is feature map size, $(h, w)$ is output position ($0 \leq h \leq H{-}1$, $0 \leq w \leq W{-}1$), $k_h, k_w$ define kernel half-width (size $(2k_h+1) \times (2k_w+1)$), $(i,j)$ are offsets over $[-k_h, k_h]$ and $[-k_w, k_w]$.
>
> In Eq. 1, substitute $m = h+i$, $n = w+j$. The original sum over $i \in [-k_h, k_h]$ covers $m \in [h - k_h, h + k_h]$. We extend to $m \in [0, H{-}1]$ safely: if $|m - h| > k_h$, then index $m{-}h{+}k_h$ falls outside $[0, 2k_h]$, i.e., outside the kernel's defined index range, so $W_c = 0$; if $m \notin [0, H{-}1]$, the input is zero-padded. Either way, added terms are zero. Same for $n$. This gives:
>
> $$Y_{b,c,h,w} = \sum_{m=0}^{H-1}\sum_{n=0}^{W-1} W_{c, m-h+k_h, n-w+k_w} \cdot X_{b,c,m,n} + b_c$$
>
> To obtain Eq. 2, linearize: $t_1 = mW + n$, $t_2 = hW + w$. Since $m, h \in [0, H{-}1]$ and $n, w \in [0, W{-}1]$, both range over $[0, HW{-}1]$, giving $\mathbf{M}_c \in \mathbb{R}^{HW \times HW}$. This matrix can represent any kernel size: for kernels larger than $(2H{-}1) \times (2W{-}1)$, excess parameters do not multiply non-zero inputs, receive no gradient, and cannot be trained; for smaller kernels, excess entries are simply zero. Define $\delta_h = m - h$, $\delta_w = n - w$:
>
> $$M_c[t_1, t_2] = \begin{cases} W_{c,\,\delta_h+k_h,\,\delta_w+k_w}, & \text{if } |\delta_h| \leq k_h \text{ and } |\delta_w| \leq k_w, \\\\ 0, & \text{otherwise.} \end{cases}$$
>
> Then $\mathbf{y}_c = \mathbf{x}_c \mathbf{M}_c + b_c$ reproduces Eq. 1, establishing Eq. 2. Crucially, $M_c[t_1, t_2]$ depends only on $(\delta_h, \delta_w)$---position pairs with the same offset share the same weight. WBMM exploits this: instead of the full $HW \times HW$ matrix, we parameterize via table $\mathbf{R}$ with $(2w_h{-}1)(2w_w{-}1)$ entries per channel (Eqs. 5--7). Full proof in revised main text.
>
> **Revision commitments:** (1) Motivation with Figure R1; (2) LKA mechanism explanation; (3) Theorem 3.2 proof in main text; (4) Numbering corrected; (5) Limitations section (see our response to Rsst).

---

> > ### Author Rebuttal · Reviewer_7ATo · 2026-04-04
> >
> > I would like to thank the authors for providing Figure R1 and detailed explanations, which helped me better grasp the main concerns they address. After re-reviewing the manuscript alongside the other reviewers' comments, I have a clearer understanding of the methodology. Despite these clarifications, I believe the current version of the manuscript still requires significant polishing. While the technical core is sound, the narrative occasionally requires a 'deep dive' that may be challenging for readers with varying levels of expertise or confidence in this specific sub-field. I first recommend that the authors integrate the intuitive explanations provided in the rebuttal directly into the main text to improve readability and accessibility. Accordingly, I intend to raise my score from 2 to 3.

---

> > > ### Author Response · Authors · 2026-04-05
> > >
> > > We sincerely thank the reviewer for recognizing that "the technical core is sound" and for the constructive suggestion to integrate the rebuttal explanations into the main text. We agree that "the narrative occasionally requires a 'deep dive' that may be challenging for readers with varying levels of expertise," and we have prepared a targeted revision to address this.
> > >
> > > **Revision overview.** All changes follow one unified thread: **WBMM replaces irregular data gathering with regular parameter traversal, and every other component is a direct consequence of this design choice.** The revised text makes this explicit at each point—GPU-level motivation grounds the irregularity problem, cross-window DW mixing addresses the limitation of non-overlapping partitions, hierarchical reparameterization combines global and local windows for multi-scale dense prediction, and inference caching exploits the input-independent weight matrix. This ensures contributions read as a coherent system rather than a collection of separate techniques. To be clear, all changes are presentational and completeness-oriented—no experimental results or technical conclusions are modified.
> > >
> > > **Specific revisions:**
> > >
> > > - **GPU-level motivation (W1) → Sec. 1, after line 065.** Figure R1 will be inserted as the new Figure 2 with a self-contained caption illustrating non-coalesced reads, cache thrashing, and O(1) FLOPs/byte intensity—grounding the abstract claim in a concrete GPU memory diagram so readers see the problem without needing systems-level background. The current Figure 2 (operator-level benchmark) will be renumbered as Figure 3, with all subsequent figures renumbered accordingly.
> > >
> > > - **LKA explanation (W2) → Sec. 2.2, replacing lines 143–152.** Expanded discussion covering what implicit GEMM with specialized CUDA kernels does (memory overhead reduction), why it remains limited (unchanged scatter-based access pattern, dispatch overhead on large maps), and quantitative evidence from Table 8.
> > >
> > > - **Theorem 3.2 proof (W4) → Sec. 3.3, replacing lines 211–219.** The complete three-step proof from our rebuttal (variable substitution with safe range extension, index linearization, offset-dependence connecting to table R) will replace the current concise statement, making the derivation fully self-contained.
> > >
> > > - **Theory-to-practice bridge → Sec. 3.3, after the proof.** A bridging paragraph explaining that the full HW×HW matrix is infeasible for practical feature maps (e.g., 56×56 yields 3136×3136), naturally motivating Windowed Batch Matrix Multiplication (Sec. 3.4) and connecting to Cross-Window Information Exchange (Sec. 3.5). This bridge is designed to prevent exactly the kind of "deep dive" the reviewer identified.
> > >
> > > - **Limitations section → after Sec. 5.** A dedicated discussion covering: (1) For windows beyond 28×28, materializing the full M becomes memory-prohibitive ($\mathbf{M} \in \mathbb{R}^{C \times d \times d}$ where $d = w^2$, so storage grows as $O(C w^4)$); scaling to larger windows requires Triton-based tiling that computes M blocks on-the-fly from the compact table R without full materialization—a planned engineering effort.
> > > (2) WBMM's core abstraction (relative position bias + batched MatMul) naturally generalizes to 1D temporal sequences and 3D video; exploring optimal window shapes and cross-window strategies in these domains is a promising future direction.
> > >
> > > - **Formatting → throughout.** All non-standard subsection numbering will be corrected per ICML template.
> > >
> > > We are grateful for the reviewer's engagement throughout this process and welcome any further suggestions.

---

### Official Review · Reviewer_KCMf · 2026-03-13

**Soundness:** 3
**Presentation:** 2
**Significance:** 3
**Originality:** 3
**Overall Recommendation:** 5
**Confidence:** 3

**Summary:**

This paper argues that large-kernel depthwise convolutions are often slow not because of FLOPs alone, but because they require gather-based access to scattered neighborhoods, which causes increasingly irregular memory access as kernel size grows. To address this, the authors use a matrix view of convolution to motivate WBMM, a windowed operator that partitions the feature map into contiguous non-overlapping blocks and applies a shared relative-position-based matrix multiplication inside each block. This keeps the convolutional inductive bias of relative-offset parameter sharing, but replaces scattered gathers with regular batched matrix multiplication, which is faster on the tested hardware. Because windowing blocks cross-window interaction, the full architecture adds lightweight 3 x 3 depthwise mixing between WBMM blocks. Experiments show that WBMM is faster than large-kernel depthwise convolution in isolated operator benchmarks, and that replacing large-kernel depthwise convolution blocks with WBMM-based blocks in real models improves training speed while preserving, and sometimes slightly improving, task performance.

**Compliance With Llm Reviewing Policy:**

Affirmed.

**Final Justification:**

I have increased my score from weak accept to accept. The paper was already technically solid and original, but my main concerns were about clarity and whether the gains depended too much on architecture-specific tuning. The rebuttal addressed these points well, especially through the controlled apples-to-apples comparison showing that replacing the spatial operator with WBMM can make the model substantially faster in at least one setting while maintaining the same performance. That result increased my confidence that the efficiency gains come from the method itself rather than from extra tuning.

**Key Questions For Authors:**

To what extent do the end-to-end gains come from WBMM itself rather than from the task-specific design choices around it? It would be helpful to report results for a more uniform replacement setting, where large-kernel convolutions are replaced with WBMM and a fixed 3 x 3 mixing schedule, without additional architecture-level tuning.

**Limitations:**

yes

**Strengths And Weaknesses:**

Strengths
- The paper identifies a specific bottleneck in large convolution kernels (irregular gather-based memory access in large-kernel depthwise convolution) and redesigns the operator around that bottleneck. The core idea of “traversing parameters instead of data” is a concrete conceptual contribution, and it leads to the distinctive empirical property that WBMM can become more efficient with larger windows, unlike standard large-kernel convolution. This is an original and well-motivated systems reframing of the large-kernel CNN problem.
- WBMM removes the expensive sliding/gather behavior of convolution, but it still preserves an important convolution-like inductive bias through its relative-position parameterization: a compact table R is expanded into the per-window matrix via a precomputed index map, so weights are shared by relative offset rather than learned as an unconstrained dense transform.
- The authors do not rely only on end-to-end model performance. They first isolate the operator in controlled benchmarks across kernel/window sizes, feature-map resolutions, and batch sizes, then validate the idea in real UniRepLKNet-style models with matched training protocols and limited modifications. Moreover, they include comparisons on various types of hardware (GPU, CPU, edge devices). This gives a fairly complete demonstration that the speedup is real at the operator level and survives in classification, segmentation, and detection settings.

Weaknesses
- Theorem 3.2 presents an exact global convolution–matrix equivalence with M in $\mathbb R^{HW \times HW}$, but Algorithm 1 then introduces a different windowed operator built on non-overlapping windows, where the per-window matrix is instead local to each window. The paper does not clearly mark where it moves from an exact statement about convolution to a restricted, hardware-motivated design choice. This makes the exposition difficult to parse and obscures the key distinction that WBMM is, as far as I understand, not an exact realization of convolution but a distinct windowed operator that is inspired by it. Without a clearer transition statement, the theorem blurs that distinction.
- Although WBMM is presented as a replacement for large-kernel depthwise convolution, the reported results appear to rely on task- and model-specific architectural tuning rather than a single recipe used uniformly across settings. In particular, the stage-wise schedule of WBMM and 3×3 mixing blocks is adjusted across implementations, and some WBMM variants introduce additional design elements such as hierarchical window reparameterization or multi-kernel fusion. As a result, the method seems to require more involved design choices to recover performance than a simple drop-in replacement for convolution, which could limit practical adoption. The main text somewhat understates this complexity when it says that the modifications are “limited to replacing large kernel convolutions with WBMM and implementing the inter-block mixing strategy.”

This is a technically solid paper with a clear and relevant motivation, an original operator design, and a convincing empirical evaluation. I see the main limitations as the clarity of the presentation and the fact that WBMM is not presented a simple drop-in replacement for convolution at the architectural level. Nevertheless, the paper advances the sub-area of efficient large-kernel CNN design in a meaningful way, and I expect its ideas to be useful for future work.

---

> ### Author Rebuttal · Authors · 2026-03-31
>
> We sincerely thank Reviewer KCMf for the thorough and constructive review.
>
> ### W1: Transition from Theorem 3.2 to Algorithm 1
>
> We fully agree this transition deserves clearer exposition. Theorem 3.2 establishes that any depthwise convolution on an $H \times W$ feature map can be exactly represented as $y_c = x_c M_c + b_c$ where $M_c \in \mathbb{R}^{HW \times HW}$. However, for large feature maps (e.g., $56 \times 56$, $HW = 3{,}136$), constructing this full matrix is impractical.
>
> **Important clarification:** WBMM is not an exact realization of global convolution, but a windowed design inspired by this equivalence---analogous to how Swin Transformer restricts self-attention to local windows. Cross-window exchange is restored through inter-block $3 \times 3$ DW convolutions (Section 3.5).
>
> **Revision:** We will add a "Design Choice" paragraph after Theorem 3.2 clearly marking the transition from exact equivalence to practical windowed approximation.
>
> ### W2/Q1: Task-specific tuning vs. drop-in replacement
>
> We conducted comprehensive controlled experiments to isolate WBMM's contribution. All configs share identical architecture/training/hyperparameters; only the spatial mixing operator differs.
>
> **Architecture note:** In all Table R1 variants, S1--S3 use inter-block mixing (spatial operator blocks alternating with $3 \times 3$ DW blocks); S4 uses all spatial operator blocks. Only the operator itself changes. "Hier" = hierarchical reparameterization ($14 \times 14$ global windows with parallel $7 \times 7$ local sub-windows, fused at inference with zero cost). For the Win7 Transformer variant, we replace WBMM blocks with non-overlapping $7 \times 7$ window self-attention blocks, setting MLP expansion ratio to 2.5 to match WBMM's parameter count and FLOPs. Training: $8 \times$ A800 GPUs. Inference speed: single A800, FP32, batch=128.
>
> **Axis 1---Controlled operator replacement (Table R1, Tiny model):**
>
> | Operator | Params(M) | FLOPs(G) | Mem(GB) | Time | Top-1 | mIoU | Spd(img/s) |
> |:---|:---:|:---:|:---:|:---:|:---:|:---:|:---:|
> | WBMM-T (w=7) | 31.0 | 4.8 | 15.16 | 6:20 | 83.2 | 48.3 | 1833.1 |
> | WBMM-T (w=14, Hier) | 33.0 | 5.1 | 15.87 | 6:31 | 83.2 | 48.8 | 1842.2 |
> | 13x13 DW + LKA | 31.0 | 5.0 | 14.77 | 6:43 | 83.1 | 48.3 | 1661.6 |
> | 13x13 DW (no LKA) | 31.0 | 5.0 | 14.77 | 9:11 | 83.1 | 48.3 | 1305.3 |
> | 13x13 DW + reparam + LKA | 31.6 | 5.1 | 17.23 | 10:21 | 83.2 | 48.7 | 1661.6 |
> | 13x13 DW + reparam (no LKA) | 31.6 | 5.1 | 17.23 | 12:46 | 83.2 | 48.7 | 1305.3 |
> | SLaK-style in WBMM design | 32.1 | 5.4 | 16.19 | 15:27 | 83.2 | 47.2 | 772.1 |
> | SLaK-Original (all stages) | 33.7 | 5.9 | 16.91 | 22:03 | 83.3 | 47.4 | 549.0 |
> | Win7 Transformer (MLP r=2.5) | 31.2 | 5.1 | 16.65 | 8:38 | 83.3 | 48.1 | 1245.1 |
>
> LKA = Large Kernel Acceleration (specialized CUDA kernels from RepLKNet/UniRepLKNet); "reparam" = UniRepLKNet structural reparameterization with kernel\_sizes=[5,7,3,3,3], dilates=[1,2,3,4,5]. SLaK-style uses parallel rectangular decomposition ($5 \times k + k \times 5 + 5 \times 5$, $k \in \{51,49,47,13\}$). Time = per-epoch ImageNet training time (min:sec).
>
> **Key observations:**
>
> (1) **WBMM w=7 vs. 13x13 DW:** Comparable classification, faster training (6:20 vs 6:43 with LKA / 9:11 without) and inference (1833 vs 1662/1305 img/s), with lower memory.
>
> (2) **WBMM w=14 Hier vs. 13x13 DW + reparam:** Complex reparameterization recovers 48.7 mIoU but at severe training cost (10:21--12:46 vs 6:31) and +1.36 GB memory. WBMM Hier still leads +0.1 mIoU with much lower overhead.
>
> (3) **WBMM vs. SLaK variants:** Despite RF=51, both SLaK variants underperform on dense prediction (47.2/47.4 vs 48.3/48.8 mIoU), 2.4--3.4x slower in training and inference.
>
> (4) **WBMM vs. Win7 Transformer:** Under matched parameters (31.0/31.2 M) and FLOPs (4.8/5.1 G), Win7 Transformer achieves comparable classification (83.3) but lower segmentation (48.1 vs 48.3/48.8 mIoU). WBMM is faster in training (6:20/6:31 vs 8:38) and inference (1833/1842 vs 1245 img/s), confirming WBMM's $O(1)$ weight construction advantage over attention's $O(B \cdot N_w)$.
>
> **Axis 2---Cross-architecture validation (ConvNeXt-T, Table R2):**
>
> | Model | Params(M) | FLOPs(G) | Mem(GB) | Top-1 | mIoU | Time |
> |:---|:---:|:---:|:---:|:---:|:---:|:---:|
> | ConvNeXt-T (7x7 DW) | 28.6 | 4.5 | 17.93 | 82.1 | 46.0 | 8:30 |
> | ConvNeXt-T -> WBMM (w=7) + 3x3 mix | 29.1 | 4.4 | 18.20 | 82.1 | 46.2 | 5:08 |
>
> Replacing all $7 \times 7$ DW with WBMM + inter-block $3 \times 3$ mixing maintains identical classification (82.1) while achieving slightly higher segmentation (46.2 vs 46.0 mIoU) with **1.66x training speedup** (5:08 vs 8:30), confirming architecture-agnostic efficiency with zero task-specific tuning.
>
> **Revision commitments:** (1) "Design Choice" paragraph after Theorem 3.2; (2) Tables R1, R2 as comprehensive ablation; (3) tighten exposition throughout.

---

> > ### Author Rebuttal · Reviewer_KCMf · 2026-04-03
> >
> > My concerns were addressed through thorough additional experiments.

---

> > > ### Author Response · Authors · 2026-04-05
> > >
> > > We thank Reviewer KCM for the careful review and for confirming that our responses have addressed the concerns. We appreciate the valuable feedback.

---

### Decision · Program_Chairs · 2026-04-30

**Decision:**

Accept (spotlight)

**Comment:**

I appreciate the review discussion and the authors’ rebuttal. On balance, I find that the paper makes a sufficiently solid case for acceptance, and the remaining concerns are not substantial enough to change that conclusion.